# INFORMING REINFORCEMENT LEARNING AGENTS BY GROUNDING LANGUAGE TO MARKOV DECISION PROCESSES

## ABSTRACT

Natural language advice has the potential to accelerate reinforcement learning, but despite significant efforts to leverage natural language for RL, utilizing *diverse* forms of language efficiently remains unsolved. Existing methods focus on mapping natural language to individual elements of MDPs such as reward functions or policies, but such approaches limit the scope of language they consider to make such mappings possible. We propose to leverage general language advice by translating sentences to a grounded formal language for expressing information about *every* element of an MDP and its solution, including policies, plans, reward functions, and transition functions. We also introduce a new model-based reinforcement learning algorithm, RLang-Dyna-Q, capable of leveraging all such advice, and demonstrate in two sets of experiments that grounding language to every element of an MDP leads to significant performance gains. In additional symbol-grounding demonstrations we show how vision-language models can annotate important structure in the environent in the form of RLang vocabulary files, eliminating the need for human labels.

## 1 INTRODUCTION

Language serves as a powerful means for humans to share information about the world, allowing us to learn more quickly or even skip learning altogether by drawing upon the domain expertise of others in the form of advice. An open question in reinforcement learning is how language advice can be leveraged to speed up learning in Markov Decision Processes (MDPs), as learning tasks *tabula rasa* is exceptionally difficult—and often impossible—in the real world. While many methods of leveraging advice for learning have emerged in the literature, a coherent theory of *language grounding* that can comprehensively support the use of language for reinforcement learning has not.

Virtually all research in language and RL grounds language to individual elements of MDPs such as policies (Liang et al., 2023; Vemprala et al., 2024; Wu et al., 2023; Andreas et al., 2017), reward functions (MacGlashan et al., 2015), and goals (Colas et al., 2020). The main drawbacks of these works is that they restrict their approach to narrow fragments of natural language. For example, the statement *"if a mug is tipped over, its contents will spill out"* clearly refers to a transition function, and mapping this information to a policy is not straightforward. For this reason, works that ground language to policies primarily focus on *imperative* sentences (e.g. *"put the pallet on the truck"*) that naturally correspond to policies, plans, or reward functions. Likewise, works that ground language to transition functions focus mainly on *declarative* sentences that provide information about the dynamics of a domain. This divergence in methodology suggests that not all language should be grounded to the same component of an MDP, and that a general language grounding system for reinforcement learning agents should be capable of grounding language to *every* element of an MDP, and its solution.

We propose a novel approach to grounding natural language for use in reinforcement learning that formulates the language grounding problem as a machine translation task from natural language to RLang (Rodriguez-Sanchez et al., 2023), a formal language designed to express information about MDPs. Our approach is akin to semantic parsing (Mooney, 2007)—a problem in natural language understanding that involves translating natural language into a formal representation—because RLang

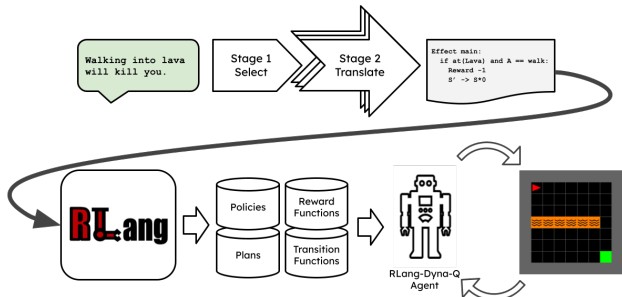

Figure 1: Translating natural language advice to RLang. We extend the original RLang pipeline (bottom) to include natural language translation and a Dyna-Q agent capable of leveraging all forms of RLang advice.

is a grounded formal language that offers a systematic means of expressing knowledge about an MDP. Such an approach calls for a learning agent capable of leveraging all such MDP components, including a partial policy, reward function, plan, and transition function. We therefore also introduce RLang-Dyna-Q, a model-based tabular RL agent based on Dyna-Q (Sutton et al., 1998), that can effectively leverage such advice. We demonstrate the strength and generality of our approach by grounding a variety of natural language advice to RLang programs, which RLang-Dyna-Q can use to significantly improve performance, sometimes making it possible to solve tasks that vanilla Dyna-Q cannot solve. Our pipeline for these experiments relies on hand-specified RLang groundings that generalize across tasks in the same domain (e.g. one grounding file for all Minigrid tasks), however, we perform demonstrations showing how these groundings can instead be partially specified by a vision-language model.

## 2 BACKGROUND

Reinforcement learning tasks are typically modeled as Markov decision processes (MDPs), which can be represented by a tuple $\langle S, A, R, T, \gamma \rangle$, where $S$ is the set of states, $A$ is the set of actions, $R$ is the reward function, $T$ is the transition function, and $\gamma$ is the discount factor. The goal of an agent is to find a policy, $\pi(a|s)$—a function that selects an action for each state—which maximizes the expected sum of discounted rewards:

$$\mathbb{E}_{\pi}\left[\sum_{t=0}^{\infty} \gamma^t R(s_t, a_t, s_{t+1})\right].$$

Value-based reinforcement learning algorithms rely on estimating the optimal action-value function $q_*$, defined as

$$q_*(s, a) = \max_{\pi} q_{\pi}(s, a),$$

providing the expected return for taking action $a$ in state $s$ and subsequently following an optimal policy (Sutton et al., 1998). Q-learning (Watkins, 1989) works to approximate $q_*$ by applying the following update rule after taking roll-outs in the environment:

$$Q(S_t, A_t) \leftarrow Q(S_t, A_t) + \alpha\left[R_{t+1} + \gamma \max_a Q(S_{t+1}, a) - Q(S_t, A_t)\right].$$

Building on Q-learning, Dyna-Q (Sutton et al., 1998) introduces an additional component: a model of the environment. While Q-learning learns from direct interaction with the environment alone, Dyna-Q builds an internal model of the environment and updates the action-value function using both real and simulated roll-outs, enabling faster convergence to the optimal action-value function.

### 2.1 LEVERAGING FORMAL SPECIFICATION LANGUAGES FOR DECISION-MAKING

Formal specification languages have long been a useful tool to inform decision-making agents. In classical planning, for example, it is standard to use the Planning Domain Description Language (PDDL; Ghallab et al. 1998) and its probabilistic extension PPDDL (probabilistic PDDL; Younes &

Table 1: Selected MDP elements, corresponding RLang groundings, and natural language interpretations. The first column shows a component of the MDP, the second shows an RLang expression about such component, and the last column contains a description of the expression.

| MDP Component | RLang Declaration | Natural Language Interpretation |
|---|---|---|
| Policy $\pi : \mathcal{S} \times \mathcal{A} \to [0, 1]$ | ```Policy build_bridge:`
`    if at_workbench:`
`        Execute use`
`    else:`
`        Execute go_to(workbench)``` | If you are at a workbench, use it. Otherwise, go to it. |
| Plan $\{A_0, A_1, ..., A_n\}$ | ```Plan gather_materials:`
`    Execute go_to(wood)`
`    Execute pickup`
`    Execute go_to(string)`
`    Execute pickup``` | Go to the wood and pick it up, then go to the string and pick it up. |
| Reward, Transition Func. $R_e : \mathcal{S} \times \mathcal{A} \times \mathcal{S} \to \mathbb{R}$ $T_e : \mathcal{S} \times \mathcal{A} \times \mathcal{S} \to [0, 1]$ | ```Effect common_sense:`
`    if at(Wall) and A == walk:`
`        Reward 0, S' -> S`
`    if at(Lava) and A == walk:`
`        Reward -1, S' -> S*0``` | Walking into walls will get you nowhere. Walking into lava will kill you. |

Littman, 2004) to specify the complete dynamics of an environment. Other languages like Linear Temporal Logic (LTL; Littman et al., 2017; Jothimurugan et al., 2019) and Policy Sketches (Andreas et al., 2017) are sufficient for describing goals and hierarchical policies, respectively, for instruction-following agents. While effective, one limiting factor of these formal languages is their narrow scope. Natural language, by contrast, can be used to express rich and varied information about nearly *all* formal elements of decision-making.

RLang (Rodriguez-Sanchez et al., 2023) is a recent formal language to emerge from the literature. While previous languages for decision-making narrowly focus on individual components of an MDP such as a policy or reward function, RLang was designed to provide information about *every* component of a structured MDP and its solution. Formally, an RLang specification is a set of RLang groundings $\mathcal{G}$ given by an RLang program $\mathcal{P}$ and an RLang vocabulary $\mathcal{V}$, a file containing a set of primitives that ground to important structured abstractions in the agent (e.g. as options, lifted skills, etc.) and the environment (e.g. as objects, factors, etc.). Importantly, vocabulary files can be designed once and reused across many tasks that occur in the same domain with little to no modifications. Some example RLang programs and their natural language interpretations can be seen in Table 1. Crucially, advice specified by RLang can be compiled directly into many components of an MDP including policies, transition functions, reward functions, and plans. Leveraging such components in a learning algorithm is not always straightforward, however, and integrating more than one component into an agent is a non-trivial problem that has not been addressed.

## 2.2 LEVERAGING NATURAL LANGUAGE FOR DECISION-MAKING

**Language in RL** Luketina et al. (2019) identify two variations of language usage in the reinforcement learning literature. The first, **language-conditional** RL, is one in which language use is a necessary component of the task. This includes environments where agents must execute commands in natural language (Mirchandani et al., 2021), or otherwise deal with language that is part of the MDP, e.g., in the observation or action space (Fulda et al., 2017; Kostka et al., 2017). The second variation is **language-assisted** RL, in which natural language is used to communicate task-related information to an agent that is *not necessary* for solving the task. In these settings, language can be used to inform policy structure (Watkins et al., 2021), reward functions (Goyal et al., 2019), transition dynamics (Narasimhan et al., 2018), or Q-functions (Branavan et al., 2012).

**Grounding Natural to Formal Languages for Planning and Learning** The notion of grounding natural language to a formal language for use in learning and planning is not new. Gopalan et al. (2018) and Berg et al. (2020) translate natural language commands into Linear Temporal Logic (LTL), which they use as reward functions for a learning agent or planning objectives, and Silver et al. (2024) and Miglani & Yorke-Smith (2020) ground natural language into PDDL, which is fed to

a recurrent neural network to output solution plans. However, the advancement of large language models (LLMs) has led to even more capable agents that for leveraging formal languages. In the planning literature, Ahn et al. (2022); Huang et al. (2022); Song et al. (2023) use primitive formal languages for executing policies on real robots or in embodied environments, Liu et al. (2023a); Xie et al. (2023) translate natural language commands into PDDL plans with the help of LLMs, and Liu et al. (2023b) proposed a modular system to ground natural language into LTL formulas. Code is also a popular choice for formal languages: Liang et al. (2023); Vemprala et al. (2024); Wu et al. (2023) use an LLM to generate Python functions as policies from natural language instructions; Singh et al. (2023) also generates programs by prompting LLMs for code completion. For learning, more recent works focus on reward design with LLMs for RL agents: Yu et al. (2023) specifies reward with LLMs through code generation and Du et al. (2023) leverage commonsense reasoning for designing reward functions. While many methods excel at grounding to formal languages Cohen et al. (2024), no existing method seeks to ground language to every component of the MDP.

### 2.3 LARGE LANGUAGE MODELS FOR MACHINE TRANSLATION

Large Language Models (LLMs), often based on architectures like the Transformer (Vaswani et al., 2017), are trained to predict the next token $x_t$ in a sequence given the preceding tokens $\{x_1, x_2, ..., x_{t-1}\}$ via the following objective:

$$\mathcal{L} = -\sum_t \log P(x_t|x_1, x_2, \ldots, x_{t-1}).$$

In very large models, this objective results in emergent capabilities such as natural language understanding and generation, making them suitable for a variety of tasks beyond mere text completion including question-answering, summarization, and more (Bubeck et al., 2023). One useful emergent capability of LLMs is the machine translation of text from one language to another. While specialized neural machine translation systems are trained using a parallel corpus to maximize the conditional probability $P(y|x)$, where $x$ is the source sequence and $y$ is the target sequence (Bahdanau et al., 2015), LLMs have achieved similar translation capabilities despite not being trained explicitly on this objective (Brown et al., 2020). Furthermore LLMs have been shown to be proficient at generating text in formal languages such as Python given a language prompt (Chen et al., 2021; Li et al., 2023).

## 3 GROUNDING NATURAL LANGUAGE ADVICE TO RLANG PROGRAMS

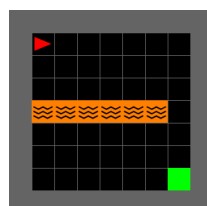 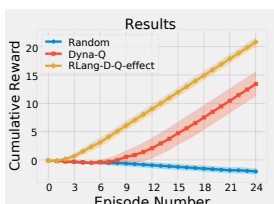 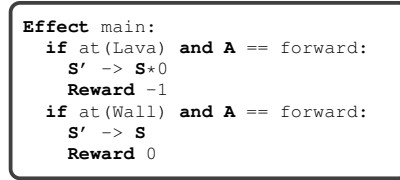

Figure 2: **LavaCrossing Experiment.** The agent was given the following advice: "Walking into lava will kill you. Walking into walls will do nothing." The initial state of LavaCrossing is pictured left, reward curves are in the center, and the grounded RLang advice is on the right.

One major motivation for leveraging language advice in RL is to supply agents with the kinds of commonsense reasoning that language can easily express. Consider the LavaCrossing environment in Figure 2. Any human interacting with this environment would quickly learn that walking into the lava squares kills you, or likewise that walking into walls will do nothing at all. Communicating this knowledge to others with language is natural for humans, but leveraging such language advice in RL is a major unsolved problem. An alternative approach to supplying commonsense advice to RL agents involves specifying it in a formal language relevant to decision-processes, which can more straightforwardly be used by a learning agent to improve learning. While such an approach is limited by the expressivity of the formal language and how it is used by the learning agent, it is fully grounded, easily interpretable, and can be extremely powerful.

As formal languages for decision-making grow more expressive, a natural next step for leveraging language advice in reinforcement learning is to translate pieces of natural language advice into

statements in such formal languages. RLang is a highly expressive candidate for language grounding because it is capable of specifying information about *every* element of a structured MDP and its solution, including plans, policies, transition functions, and reward functions (see Table 2 in Rodriguez-Sanchez et al. (2023)). Furthermore, we hypothesize that different kinds of advice can most naturally be represented by different components of the MDP, and that methods that ground language to a single component are insufficient to capture general language advice. For example, the statement, *"stacked dishes can topple if unevenly piled,"* is precisely a statement about transition dynamics, and while it can ultimately be used to inform a plan or policy, the information contained in the statement would not be retrievable if it were not represented as a partial transition function; the most harmonious representation of the advice is as a partial transition function. Likewise, the sentence, *"wear oven mitts whenever handling pots and pans,"* is a statement about a policy, and representing it as a reward function would only indirectly capture its meaning.

We therefore formulate the language grounding problem in RL as a machine translation task from natural language to RLang. Our task is as follows: given an RLang vocabulary $\mathcal{V}$—a set of task-general groundings that act as primitives in an RLang program—for a given MDP and a piece of natural language advice $u$, we seek a function $\phi : u \times \mathcal{V} \to \mathcal{P}_u$, where $\mathcal{P}_u$ is an executable RLang program capturing the advice in $u$ that can be leveraged by a learning agent. We propose to do this translation entirely in-context using a general-purpose large language model in a two-stage pipeline by 1) identifying which RLang grounding type would best capture the language advice; and 2) translating the advice into an RLang program. Stage 1, the selection stage, instructs the LLM to classify a novel piece of advice $u$ into RLang grounding types such as Effects, Policies, and Plans, consulting a small number of example classifications in the prompt. This ensures that the advice will be represented by an appropriate component of the MDP.[1] Stage 2, the translation stage, instructs the LLM to translate $u$ to an RLang program specifying the grounding type given by Stage 1 using roughly 5 example translations in the prompt that were hand-engineered to cover a wide range of RLang's syntax. These programs are compiled using RLang's compiler into Python functions corresponding to transition functions, reward functions, policies, and plans that can be leveraged by a learning agent. In experiments we demonstrate that this pipeline effectively grounds the advice to useful MDP components. Our pipeline is illustrated in Figure 1.

### 3.1 RLANG-DYNA-Q: A SINGLE AGENT FOR LEVERAGING ALL OF RLANG

In the original RLang paper, the authors presented a number of RLang-enabled agents—including ones based on Q-Learning, PPO (Schulman et al., 2017), and DOORmax (Diuk et al., 2008)—each capable of leveraging *individual* RLang groundings to improve learning. However, leveraging general language advice requires integrating potentially *all* RLang groundings into a single learning agent. We therefore introduce RLang-Dyna-Q, a learning agent based on Dyna-Q (Sutton et al., 1998) that is capable of simultaneously leveraging a partial policy, plan, reward function, and transition function given by an RLang program. Similar to Dyna-Q, RLang-Dyna-Q leverages the Bellman update rule to update Q-values using rollouts collected both from environment interaction and from simulated interaction, which is generated from a partial model of the environment that is learned over time. However, RLang-Dyna-Q also leverages a partial model given by an RLang program to generate simulated rollouts before learning begins (see Algorithm 1, our modifications to Dyna-Q are in blue). Dyna-Q is an appropriate core learning agent because integrating actions and dynamics is most natural in a model-based learning algorithm that explicitly represents a policy, transition function, and reward function.

## 4 EXPERIMENTS

We hypothesize that RLang is an effective grounding for natural language advice in the context of reinforcement learning. However, evaluating whether language advice $u$ and RLang program $\mathcal{P}_u$ have the same semantic content is difficult, so we designed our experiments to test the objective of primary interest: the agent's performance on a learning task. If we provide advice that is helpful to the agent, then grounding it properly should improve performance. We therefore assessed our translation pipeline by evaluating agent performance on multiple custom tasks based on the Minigrid/BabyAI

---

[1]We assume that each piece of advice—which may contain multiple sentences—grounds to a single RLang grounding type. This constraint can easily be relaxed in future work.

---

**Algorithm 1** RLang-Dyna-Q Agent

**Given:** $\pi_{\text{RLang}}, T_{\text{RLang}}, R_{\text{RLang}}$ from an RLang program
Init $Q(s,a), T(s,a), R(s,a)$ for all $s \in \mathcal{S}, a \in \mathcal{A}(s)$
**loop**
    $s \leftarrow$ current (nonterminal) state
    $a \leftarrow \epsilon_1, \epsilon_2$-greedy$(s, \pi_{\text{RLang}}, Q)$        # With prob. $\epsilon_2$, we execute the RLang plan or policy
    Execute action $a$; observe next state $s'$, and reward $r$
    $Q \leftarrow Q(s,a) + \alpha\left[r + \gamma \max_{a'} Q(s',a') - Q(s,a)\right]$
    $T(s,a), R(s,a) \leftarrow s', r$     # Update our model
    **for** $i = 1$ to $N_1$ **do**
        $s \leftarrow$ random previously observed state
        $a \leftarrow$ random action previously taken in $s$
        $s', r \leftarrow T(s,a), R(s,a)$
        $Q \leftarrow Q(s,a) + \alpha\left[r + \gamma \max_{a'} Q(s',a') - Q(s,a)\right]$
    **end for**
    **for** $i = 1$ to $N_2$ **do**
        $s \leftarrow$ random previously observed state
        $a \leftarrow$ random action **not** previously taken in $s$
        $s', r \leftarrow T_{\text{RLang}}(s,a), R_{\text{RLang}}(s,a)$       Predict $s', r$ using dynamics given by RLang
        $Q \leftarrow Q(s,a) + \alpha\left[r + \gamma \max_{a'} Q(s',a') - Q(s,a)\right]$
    **end for**
**end loop**

---

(Chevalier-Boisvert et al., 2023; Chevalier-Boisvert et al., 2019) and VirtualHome (Puig et al., 2018) environments given expert advice. We include ablations of different RLang components to demonstrate our auxiliary hypothesis, that language is best grounded to *every* element of an MDP. Additionally, we run a small user study to assess our pipeline's efficacy on a wide range of non-expert language advice. Finally, we perform a series of symbol-grounding demonstrations showing how vision-language models can eliminate the need for human-readable RLang vocabulary files, which require human labeling.

## 4.1 LEVERAGING EXPERT ADVICE IN MINIGRID

We designed custom environments using the Minigrid/BabyAI library, a platform for studying the behavior of language-informed agents. In a typical Minigrid environment, an agent might reason about opening and closing doors using keys which may be hidden in other rooms, managing a small inventory of items, removing obstacles like balls out of the way to reach other rooms or objects, and avoiding lava, all for the ultimate purpose of reaching a goal. Minigrid environments are an ideal setting for our experiments for three reasons: 1) they can be solved using tabular RL algorithms, which our informed, model-based RLang-Dyna-Q agent is based on; 2) there are clear and obvious referents of language in both the state and action spaces of these environments (e.g. keys, doors, and balls are represented neatly in a discrete state space and skills such as walking towards objects are easy to implement); 3) many objects are shared across environments enabling the reuse of a common RLang vocabulary for referencing these objects, which makes it easier for our translation pipeline to ground novel advice.

We provide a domain-general RLang vocabulary file $\mathcal{V}$ containing a set of RLang groundings to be used as primitives in a full RLang program. These vocabulary files are generated automatically for each minigrid environment given a single general template, and include perception abstractions such as the objects in the environment (e.g., `yellow_key`, `red_door`) and a short list of predicates for reasoning with them (e.g., `carrying()`, `reachable()`, `at()`), as well as a single abstract action in the form of a lifted skill for walking to any reachable object (`go_to()`). Importantly, these groundings have semantically-meaningful labels, which enable a simple translation process.[2] All agents in the experiments, including the Random, Dyna-Q, and RLang-Dyna-Q agents, are

---

[2]In our final demonstrations, we relax this constraint by determining the semantic label of entity groundings and utilizing images of the objects in the environment to ground the referents of ambiguous advice using an off-the-shelf vision-language model.

given access to this lifted skill. However, we do not provide the Dyna-Q agent with any perception abstractions, as including them induces an equivalent state space in the tabular RL setting. Likewise, the Random agent does not consider state when selecting an action. In Stage 2 of the translation pipeline, we provide the list of available RLang groundings that can be referenced in an RLang program along with the language advice. This prevents the LLM from hallucinating imaginary skills, objects, and predicates when translating the advice into an RLang program. The LLM never interacts with the MDP directly. The translation examples used in the prompts in both stages of translation did not change across experiments, though these translations are vocabulary-specific and grounding advice to environments outside of Minigrid will require domain-compatible example translations.

We evaluated our grounding pipeline on four diverse Minigrid environments: LavaCrossing, Multi-Room, MidMazeLava, and HardMaze. For each environment, we collected multiple pieces of natural language advice from human experts—people familiar with both the environment and how the agent interacts with it via perception and action, i.e. the skills the agent has access to and the fact that its perception consists of objects and simple predicates—and translated them into RLang programs using our two-stage pipeline. Each piece of advice was translated to a single RLang grounding type, and each piece of advice contained multiple sentences. We then evaluated our RLang-Dyna-Q agent on each environment with the translated RLang programs. In the MidMazeLave environment, advice was grounded to multiple RLang types. We include additional results for RLang-Dyna-Q utilizing only one type of advice at a time—Effects, Plans, or Policies—to isolate their impact on performance.

RLang-Dyna-Q significantly outperformed vanilla Dyna-Q in all of the experiments. In LavaCrossing (see Figure 2), the agent is tasked with reaching a goal while avoiding lava, and merely advising the agent about the dangers of lava and futility of walking into walls greatly increases performance. In the MultiRoom environment (see Figure 9), in which the agent must open a series of doors to reach a goal, providing a plan in natural language significantly increased performance. In MidMazeLava (see Figure 3) and HardMaze (see Figure 4), the agent is faced with significantly more difficult tasks. In the former, the agent must unblock doors and open them with keys to reach a goal while avoiding lava, and in the latter the agent must traverse through many rooms, bringing keys across rooms to doors which must be unblocked to reach a goal. We collected paragraphs-worth of advice for these environments, which we translated into RLang plans, policies, and effects. In HardMaze, this language advice made it possible to solve the task, as the vanilla Dyna-Q agent did no better than random. For each experiment, 10 instances of each agent were run to generate a 95% confidence interval on their cumulative reward over 50 episodes (LavaCrossing was run for 25 episodes only). The number of timesteps per episode varied across environments.

## 4.2 LEVERAGING EXPERT ADVICE IN VIRTUALHOME

We ran additional experiments on custom environments based on the VirtualHome library, a platform for simulating complex household activities. VirtualHome has an object-oriented state space, which can be referenced natively in RLang. We engineered 2 tasks in a kitchen environment to assess our language grounding pipeline: FoodSafety (see Figure 5), where the agent is tasked with putting a pie into the fridge and salmon into the microwave, and CouchPotato, where the agent is tasked with bringing a remote control to a sofa and putting cereal into a kitchen cabinet, while avoiding picking up toothpaste. In these environments, agents are given an RLang vocabulary file with groundings for object-oriented perception and action abstractions such as the objects in the environment (e.g. `salmon_327`, `fridge_305`), a short list of predicates (e.g. `inside()`, `holding()`, `near()`), and a set of lifted skills (e.g. `walk_to()`, `open`, `grab`). These groundings have semantically meaningful labels, which make them easy targets for grounding natural language.

Our experimental design here is identical to the Minigrid experiments: for each environment we collected multiple pieces of language advice from human experts and translated them into RLang programs via our two-stage pipeline. We then evaluated the performance of an RLang-Dyna-Q agent on our environments in comparison to a vanilla Dyna-Q agent. In all of our experiments, the RLang-informed agents significantly outperformed Dyna-Q. For each experiment, 10 instances of each agent were run to generate a 95% confidence interval on their cumulative reward over 50 and 70 episodes for FoodSafety and CouchPotato, respectively. The maximum number of timesteps per episode varied. The agent parameters are listed in the appendix.

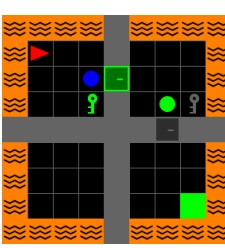
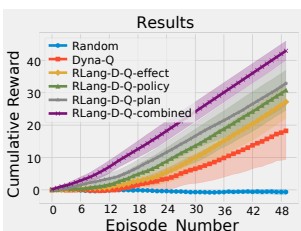

"Pick up the blue ball and drop it to your right. Then pick up the green key and unlock the green door. Then drop the key to your right." "Some general advice: If you are carrying a key and its corresponding door is closed, open the door if you are at it, otherwise go to the door if you can reach it. Otherwise, drop any keys for doors you can't reach. If you can reach the goal, go to it." "Walking into lava will kill you. If you're not at a door, toggling will do nothing. Trying to pick something up while you're carrying something is pointless. Walking into walls will do nothing."

Figure 3: **MidMazeLava Experiment.** Language advice given to the agent was grounded to RLang effects, plans, and policies. The full translated RLang program is available in the appendix. All RLang-Dyna-Q agents outperformed Dyna-Q.


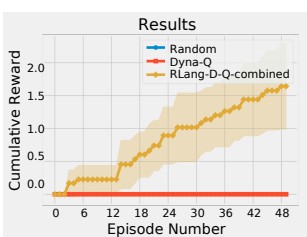

"Go and pick up the green ball, and drop it on your left, and then go pick up the blue key, and go to the blue door and open it up and drop the key on your left, and then go pick up the green key, and go to the green door to open it and drop the key on your left, and then go pick up the purple ball and drop it on your right." "Nothing will happen if you walk towards the wall, or try to open a purple door without the purple key if it is locked. The applies for the yellow door and key as well as the red door and key." "If you can reach the grey door and it is closed but you have the key, open it if you are at it or otherwise go to it. The same applies to the purple door, yellow door, and red door. Lastly, if you find the goal is reachable just go to the goal directly."

Figure 4: **HardMaze Experiment.** Language advice given to the agent was grounded to RLang effects, plans, and policies. The full translated RLang program is available in the appendix. Vanilla Dyna-Q was not able to complete this task.

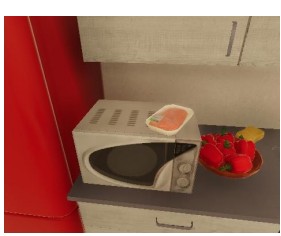
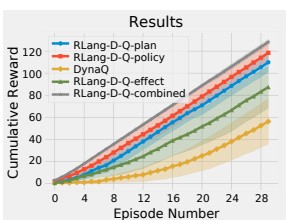

"Go to fridge and open it, and then go find the pie and pick it up, walk back to the fridge and put the pie in the fridge. You have to close the fridge too", "If the salmon is in the microwave, and you are at the microwave and it's open, close it. Otherwise if you are holding salmon, do the following: open the microwave if you are near it but it's closed, put the salmon into the microwave if it's open and you're near it, else walk to the microwave.", "If the pie is in the fridge, and the salmon is in the microwave, then closing the fridge if the microwave is closed or closing the microwave if the fridge is closed will give you reward and end the episode."

Figure 5: **FoodSafety Experiment.** Language advice given to the agent was grounded to RLang effects, plans, and policies. The full translated RLang program is in the appendix. All RLang-Dyna-Q agents outperformed Dyna-Q.

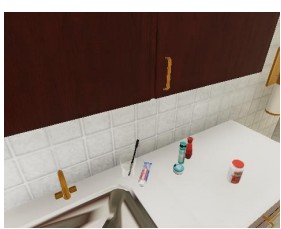
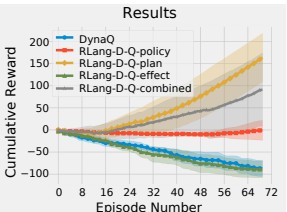

"If you're holding the toothpaste and can drop it, drop it.", "Go grab the remote control and put it on the sofa.", "If you're holding the toothpaste and are not trying to drop it, you will be penalized. Also, nothing will happen if you try to walk to the remote control, cereal, toothpaste, or salmon, if you try to walk to them and they are contained inside anything."

Figure 6: **CouchPotato Experiment.** Language advice given to the agent was grounded to RLang effects, plans, and policies. All the RLang agents outperformed Dyna-Q with the exception of the Effect-enabled agent. We note that bugs in the simulator non-deterministically prevent certain actions from executing, so the advice specified only applies part of the time, leading to decreased performance.

The impact of each kind of advice (e.g. plans, policies, transitions, and rewards) varied across tasks in the VirtualHome and Minigrid experiments, with some environments benefiting primarily from plan-centric advice and others benefiting most from policy advice. In virtually all cases, model-centric advice—about transitions and rewards—was less valuable than other forms of advice. We suggest that this discrepancy is due to how useful model-based advice is in comparison to explicit policy and planning advice. While policy and planning advice describe which actions to take in a given context, model-based advice was often used to suggest which actions *not* to take, relying on the underlying learning agent to find the best action. Furthermore, model-based advice was useful less of the time, i.e. in fewer states. This is best illustrated by comparing the relative performance of effect-enabled RLang-Dyna-Q agents with policy and plan-enabled agents in the MidMazeLava Experiment in Figure 3 and the FoodSafety Experiment in Figure 5. The model-based advice in the first experiment is to avoid lava, which there are many opportunities to walk into, resulting in the performance of the effect-enabled agent closer to the plan and policy-enabled agents. By comparison, the model-based advice in the second experiment is more niche, accounting only for a handful of transitions, and the effect-enabled agent correspondingly performs closer to baseline Dyna-Q than to the plan and policy-enabled agents.

### 4.3 Grounding Symbols with a Vision-Language Model

A crucial assumption made by our pipeline is that we are given semantically-meaningful labels for the groundings we have, including labels for objects (e.g. `salmon`), skills (e.g. `go_to(kitchen)`), and truth-valued predicates (e.g. `is_open(fridge)`). Assigning relevant labels for these groundings enables a relatively simple translation from natural language into RLang. In a real-world setting, we imagine that the labels for these groundings can be generated in two ways: 1) prescriptively, in the case of skill engineering by humans, and 2) via a pre-trained foundation model for identifying predicates and objects in the environment. Implementing a full symbol-grounding system[3] is outside the scope of this work, however, we performed an additional demonstration showing how the labels for object groundings could be easily extracted from images of the VirtualHome environments using a vision-language model.

In addition to performing the initial semantic labeling of objects, we demonstrated how incorporating a vision-language model in-the-loop could expand the variety of advice our system is capable of grounding. By asking a VLM to disambiguate referents using images of entities in the environment, we are able to successfully ground entities for semantically-ambiguous advice. Given 11 images of the entities in the VirtualHome environment, we asked GPT-4o to ground the referents of 17 ambiguous commands that would require visual and operational knowledge of the entities in the scene. For example, we can ground noun phrases like "the white box you might put food in" to a white microwave in the scene or "the tall red box" to a tall red refrigerator (see Figure 7). These additional experiments—the automatic semantic labeling of entities in the environment and the in-the-loop semantic grounding of ambiguous referents in advice—ameliorate the need for expert-crafted, semantically perfect RLang grounding files.

### 4.4 Evaluating Translation and RLang Efficacy

To assess RLang's ability to capture the breadth of general language advice, we ran a small user study. We asked 10 undergraduate students to solve the LockedRoom MiniGrid task (pictured in Figure 8) and then asked them to describe in one or two sentences any advice they would give to an agent completing the task for the first time. We collected their responses and ran them through our translation pipeline to arrive at the RLang groundings in Table 3 of the Appendix. Of 10 pieces of advice collected, 9 were translated into valid RLang programs, while 1 referenced groundings that did not exist (e.g. `second_left_door`). We used the remaining valid RLang programs to inform 9 separate RLang-Dyna-Q agents that we compared against a baseline Dyna-Q agent given no advice. With a few exceptions, providing advice either did not meaningfully impact performance over the baseline or led to dramatic improvements in performance (see Table 2). In the cases where advice did not impact performance, it was translated into a parsable RLang program that referenced groundings that were not in the RLang vocabulary file (e.g. "the second left door" was translated to `second_left_door`, but a proper reference would be `yellow_door`). We address this symbol-

---

[3]Learning a mapping from symbolic labels to groundings is explored in Steels & Hild (2012).

**Visual perception:**

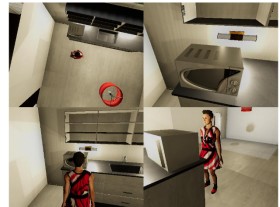

**Query:**
*"What is the object that the character is looking at?"*

**Response:**
*"microwave"*

**Visual perception:**

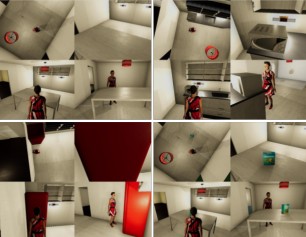

**Advice:**
*"Put the blue box in the red box."*

**Grounding:**
*"'blue box' refers to cereal 'red box' refers to fridge"*

```
Class Cereal:
   object_id: int
Class Fridge:
   object_id: int
Class Microwave:
   object_id: int

Object cereal_235 :=
     Cereal(235)
Object fridge_122 :=
     Fridge(122)
Object microwave_98 :=
     Microwave(98)
```

Figure 7: By capturing images of unnamed objects in the environment specified only by numeric object ids we can prompt a VLM to provide us with semantic labels. The labels output by VLM can be provided to the translation pipeline as semantic primitives (see right). Additionally, we demonstrate how the VLM can be re-prompted after an initial labeling to resolve semantic ambiguities that require visual knowledge of the entities in question. After disambiguation, the advice can be re-written to incorporate the true grounding labels (e.g. `cereal_235` instead of "blue box"), and then processed by the remaining grounding pipeline. Additional grounding examples are reported in the appendix.

grounding failure in the VirtualHome environment by using a VLM to ground semantically-ambiguous referents (see section 4.3). Failures also occurred when users specified plans whose pre-conditions were not met at the start state of the environment and failed to execute (e.g. the last piece of advice suggests to go to the room with the red key, but the agent cannot visit the room without first opening the grey door).

## 5 DISCUSSION AND CONCLUSION

Natural language grounding (Steels & Hild, 2012) has critical implications for all of AI. Just as RL is intended as a model of intelligent decision making, we propose that its core formalisms offer a natural target for language grounding. If MDPs model human decision-making, and humans invented language to share information that aids their decision-making, then the appropriate target for language grounding should be an MDP, or a richer and perhaps more structured decision process reflecting the complexity of human decision-making. One line of evidence for this claim is the direct correspondence between parts of speech and elements of structured decision-processes (Rodriguez-Sanchez et al., 2020). For example, the object classes in Object Oriented MDPs (Diuk et al., 2008) naturally correspond to the concept of **common nouns** requiring **determiners** to single out class instances, and the parameters in Parameterized Action MDPs Masson et al. (2016) naturally correspond to **adverbs** for modifying the execution of discrete macro-actions (**verbs**).

More practically, knowledge expressed in natural language has immense potential to inform reinforcement learning agents, and thereby alleviate the high sample complexity of having to learn *tabula rasa*. We present a novel method for leveraging general natural language advice to expedite learning in Markov Decision Processes by translating it into RLang, a formal language designed to specify information about every element of an MDP and its solution. Our method can ground advice to reward functions, transition functions, plans, and policies. We also introduce a modified Dyna-Q agent capable of leveraging all of the types of information present in the partial MDP specification represented by RLang. Our findings show that our approach can leverage a wide variety of language advice to accelerate learning.

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

# A APPENDIX

## A.1 USER STUDY

Table 2: **User Study.** We collected 10 pieces of advice from 10 undergraduate students for the LockedRoom environment. For each piece of advice, 5 agent instances were run for 25 episodes on the LockedRoom environment for 500 steps. The cumulative discounted reward for the 25 episodes is in the first column along with a 95% confidence interval. The average percent increase in cumulative discounted reward over the baseline is present in the second column. The second-to-last piece of advice did not ground to a valid RLang program, so no experiment was run.

| Avg Cumulative Return | % improvement | Natural Language Advice |
|---|---|---|
| $17.86 \pm 2.36$ | — | No advice |
| $22.01 \pm 0.71$ | $+23.24$ | "Remember to toggle to open doors." |
| $16.79 \pm 1.75$ | $-5.99$ | "You don't need to carry keys to open the grey door." |
| $17.22 \pm 1.75$ | $-3.60$ | "Identify the room with the red key, move to that room by opening the door. Pick up the key. Identify the room with the red door, proceed there. Open the red door. Find the green square and go there to finish the game." |
| $23.55 \pm 0.69$ | $+31.86$ | "Move to the grey door, open it and enter the room until you get to the red key, pick it up. Exit the room and move towards the red door, open it and get into that room. Move to the green block and enter it." |
| $23.93 \pm 0.37$ | $+33.99$ | "Go to the grey door. open the grey door. go to the red key. pick up the red key. go to the red door. open the red door. go to the green square." |
| $24.07 \pm 0.22$ | $+34.77$ | "Pick up the red key after opening the grey door. Then walk to the red door, open it, and go to the goal." |
| $17.57 \pm 0.78$ | $-1.63$ | "You cannot open the red door without a red key." |
| $17.77 \pm 0.54$ | $-0.50$ | "Walking towards the red door is not very useful if it is closed." |
| — | — | "Go down until the second door on the left and pick up the key. Then exit the room and go down until the next door on the left and use it to open the door and get to the green box." |
| $18.35 \pm 1.93$ | $+2.76$ | "Go to the room that has the red key, pick it up, and then go to the room with a red door. Enter the room, and go to the green goal object." |

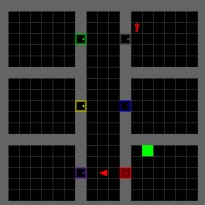

Figure 8: The initial state of the LockedRoom environment.

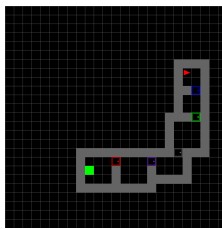 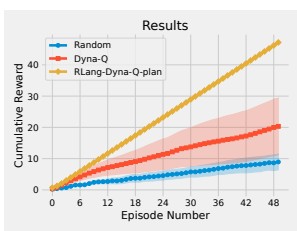 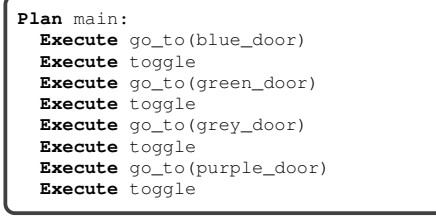

Figure 9: **MultiRoom Experiment.** The agent was given the following advice: "First go to the blue door, then the green door, then the grey door, then the purple door." The initial state of MultiRoom is pictured on the left, reward curves are center, and the translated advice is on the right.

### A.2 PROMPTS USED FOR TRANSLATION PIPELINE FOR MINIGRID EXPERIMENTS

#### A.2.1 [MINIGRID] PROMPT USED FOR STAGE 1 OF THE TRANSLATION PIPELINE. GIVEN A NEW PIECE OF ADVICE, WE PROMPT THE LLM TO CLASSIFY IT AS AN EFFECT, PLAN, OR POLICY.

RLang is a formal language for specifying information about every element of a Markov Decision Process (S,A,R,T). Each RLang object refers to one or more elements of an MDP. Here is a description of three important RLang groundings:

Policy: a direct function from states to actions, best used for more general commands.
Effect: a prediction about the state of the world or the reward function.
Plan: a sequence of specific steps to take.

Your task is to decide which RLang grounding most naturally corresponds to a given piece of advice:
Advice = "Don't touch any mice unless you have gloves on."
Grounding: Effect
Advice = "Walking into lava will kill you."
Grounding: Effect
Advice = "First get the money, then go to the green square."
Grounding: Plan
Advice = "Go through the door to the goal."
Grounding: Plan
Advice = "If you have the key, go to the door, otherwise you need to get the key."
Grounding: Policy
Advice = "If there are any closed doors, open them."
Grounding: Policy

### A.2.2 [MINIGRID] PROMPT USED FOR STAGE 2 OF THE PIPELINE TO TRANSLATE A PIECE OF ADVICE INTO AN RLANG PLAN.

Your task is to translate natural language advice to RLang plan, which is a sequence of specific steps to take. For each instance, we provide a piece of advice in natural language, a list of allowed primitives, and you should complete the instance by filling the missing plan function. Don't use any primitive outside the provided primitive list corresponding to each instance, e.g., if there is no 'green_door' in the primitive list you must not use 'green_door' for the plan function.

Advice = "Open the door with the key and go through it to the goal"
Primitives = ['Agent', 'Wall', 'GoalTile', 'Lava', 'Key', 'Door', 'Box', 'Ball', 'left', 'right', 'forward', 'pickup', 'drop', 'toggle', 'done', 'pointing_right', 'pointing_down', 'pointing_left', 'pointing_up', 'go_to', 'step_towards', 'yellow_key', 'yellow_door', 'agent', 'goal', 'at', 'in_inventory']

```
Plan main:
    Execute go_to(yellow_key)
    Execute pickup
    Execute go_to(yellow_door)
    Execute toggle
    Execute go_to(goal)
```

Advice = "Get the key behind the red door to open the grey door. Then drop the key to the left."
Primitives = ['Agent', 'Wall', 'GoalTile', 'Lava', 'Key', 'Door', 'Box', 'Ball', 'left', 'right', 'forward', 'pickup', 'drop', 'toggle', 'done', 'pointing_right', 'pointing_down', 'pointing_left', 'pointing_up', 'go_to', 'step_towards', 'yellow_key', 'yellow_door', 'agent', 'goal', 'at', 'in_inventory']

```
Plan main:
    Execute go_to(red_door)
    Execute toggle
    Execute go_to(grey_key)
    Execute pickup
    Execute go_to(grey_door)
    Execute toggle
    Execute left
    Execute drop
```

### A.2.3 [Minigrid] Prompt used for Stage 2 of the pipeline to translate a piece of advice into an RLang policy.

Your task is to translate natural language advice to RLang policy, which is a direct function from states to actions. For each instance, we provide a piece of advice in natural language, a list of allowed primitives, and you should complete the instance by filling the missing policy function. Don't use any primitive outside the provided primitive list corresponding to each instance, e.g., if there is no 'green_door' in the primitive list you must not use "green_door' for the policy function.

Advice = "If the yellow door is open, go through it and walk to the goal. Otherwise open the yellow door if you have the key."
Primitives = ['Agent', 'Wall', 'GoalTile', 'Lava', 'Key', 'Door', 'Box', 'Ball', 'left', 'right', 'forward', 'pickup', 'drop', 'toggle', 'done', 'pointing_right', 'pointing_down', 'pointing_left', 'pointing_up', 'go_to', 'step_towards', 'yellow_key', 'yellow_door', 'agent', 'goal', 'at', 'carrying']

```
Policy main:
  if yellow_door.is_open:
    Execute go_to(goal)
  elif carrying(yellow_key) and at(yellow_door) and not yellow_door.is_open:
    Execute toggle
```

Advice = "If you don't have the key, go get it."
Primitives = ['Agent', 'Wall', 'GoalTile', 'Lava', 'Key', 'Door', 'Box', 'Ball', 'left', 'right', 'forward', 'pickup', 'drop', 'toggle', 'done', 'pointing_right', 'pointing_down', 'pointing_left', 'pointing_up', 'go_to', 'step_towards', 'grey_key', 'red_door', 'grey_door', 'agent', 'purple_ball', 'at', 'carrying']

```
Policy main:
  if at(grey_key):
    Execute pickup
  elif not carrying(grey_key):
    Execute go_to(grey_key)
```

Advice = "If you are carrying a ball and its corresponding box is closed, open the box if you are at it, otherwise go to the box if you can reach it."
Primitives = ['Agent', 'Wall', 'GoalTile', 'Lava', 'Key', 'Door', 'Box', 'Ball', 'left', 'right', 'forward', 'pickup', 'drop', 'toggle', 'done', 'pointing_right', 'pointing_down', 'pointing_left', 'pointing_up', 'go_to', 'step_towards', 'green_ball', 'green_box', 'purple_box', 'agent', 'purple_ball', 'at', 'reachable', 'carrying']

```
Policy main:
  if carrying(green_ball) and not green_box.is_open:
    if at(green_box):
      Execute toggle
    elif reachable(green_box):
      Execute go_to(green_box)
```

Advice = "Drop any balls for boxes you can't reach"
Primitives = ['Agent', 'Wall', 'GoalTile', 'Lava', 'Key', 'Door', 'Box', 'Ball', 'left', 'right', 'forward', 'pickup', 'drop', 'toggle', 'done', 'pointing_right', 'pointing_down', 'pointing_left', 'pointing_up', 'go_to', 'step_towards', 'green_ball', 'green_box', 'purple_box', 'agent', 'purple_ball', 'at', 'reachable', 'carrying']

```
Policy main:
  if carrying(green_ball) and not reachable(green_box):
    Execute drop
  if carrying(purple_ball) and not reachable(purple_box):
    Execute drop
```

Advice = "if you have any key for a door that you cannot reach, you should drop it"
Primitives = ['Agent', 'Wall', 'GoalTile', 'Lava', 'Key', 'Door', 'Box', 'Ball', 'left', 'right', 'forward', 'pickup', 'drop', 'toggle', 'done', 'pointing_right', 'pointing_down', 'pointing_left', 'pointing_up', 'go_to', 'step_towards', 'green_ball', 'green_box', 'purple_box', 'agent', 'purple_ball', 'at', 'reachable', 'carrying']

```
Policy main:
  if carrying(green_key) and not reachable(green_door):
    Execute drop
  if carrying(purple_key) and not reachable(purple_door):
    Execute drop
  if carrying(red_key) and not reachable(red_door):
    Execute drop
```

Advice = "Hey listen, you can open the door if you have the key and at the door when the door is closed"
Primitives = ['Agent', 'Wall', 'GoalTile', 'Lava', 'Key', 'Door', 'Box', 'Ball', 'left', 'right', 'forward', 'pickup', 'drop', 'toggle', 'done', 'pointing_right', 'pointing_down', 'pointing_left', 'pointing_up', 'go_to', 'step_towards', 'green_ball', 'green_box', 'purple_box', 'agent', 'purple_ball', 'at', 'reachable', 'carrying']

```
Policy main:
  if carrying(purple_key) and not purple_door.is_open and at(purple_door):
    Execute toggle
```

1026
1027
1028
1029
1030
1031
1032
1033
1034
1035
1036
1037
1038
1039
1040
1041
1042
1043
1044
1045
1046
1047
1048
1049
1050
1051
1052
1053
1054
1055
1056
1057
1058
1059
1060
1061
1062
1063
1064
1065
1066
1067
1068
1069
1070
1071
1072
1073
1074
1075
1076
1077
1078
1079

### A.2.4 [MINIGRID] PROMPT USED FOR STAGE 2 OF THE PIPELINE TO TRANSLATE A PIECE OF ADVICE INTO AN RLANG EFFECT.

Your task is to translate natural language advice to RLang effect, which is a prediction about the state of the world or the reward function. For each instance, we provide a piece of advice in natural language, a list of allowed primitives, and you should complete the instance by filling the missing effect function. Don't use any primitive outside the provided primitive list corresponding to each instance, e.g., if there is no 'green_door' in the primitive list you must not use 'green_door' for the effect function.

Advice = "Don't go to the door without the key"
Primitives = ['yellow_door', 'goal', 'pickup', 'yellow_key', 'toggle', 'go_to', 'carrying', 'at']

```
Effect main:
  if at(yellow_door) and not carrying(yellow_key):
    Reward -1
```

Advice = "Don't walk into closed doors. If you're tired, don't go forward."
Primitives = ['Agent', 'Wall', 'GoalTile', 'Lava', 'Key', 'Door', 'Box', 'Ball', 'left', 'right', 'forward', 'pickup', 'drop', 'toggle', 'done', 'pointing_right', 'pointing_down', 'pointing_left', 'pointing_up', 'go_to', 'step_towards', 'green_ball', 'green_box', 'purple_box', 'agent', 'purple_ball', 'at', 'reachable', 'carrying']

```
Effect main:
  if at(yellow_door) and yellow_door.is_closed and A == forward:
    Reward -1
    S' -> S
  elif tired() and A == forward:
    Reward -1
```

Advice = "Walking into balls is pointless. You will die if you walk into keys. Trying to open a box when you aren't near it will do nothing."
Primitives = ['Agent', 'Wall', 'GoalTile', 'Lava', 'Key', 'Door', 'Box', 'Ball', 'left', 'right', 'forward', 'pickup', 'drop', 'toggle', 'done', 'pointing_right', 'pointing_down', 'pointing_left', 'pointing_up', 'go_to', 'step_towards', 'green_ball', 'green_box', 'purple_box', 'agent', 'purple_ball', 'at', 'reachable', 'carrying']

```
Effect main:
  if at(Ball) and A == forward:
    Reward 0
    S' -> S
  elif at(Key) and A == forward:
    Reward -1
    S' -> S*0
  elif at(Box) and A == toggle:
    Reward 0
    S' -> S
```

## A.3 Prompts Used for Translation Pipeline for VirtualHome Experiments

### A.3.1 [VirtualHome] Prompt used for Stage 1 of the translation pipeline. Given a new piece of advice, we prompt the LLM to classify it as an Effect, Plan, or Policy.

RLang is a formal language for specifying information about every element of a Markov Decision Process (S,A,R,T). Each RLang object refers to one or more elements of an MDP. Here is a description of three important RLang groundings:

Policy: a direct function from states to actions, best used for more general commands.
Effect: a prediction about the state of the world or the reward function.
Plan: a sequence of specific steps to take.

Your task is to decide which RLang grounding most naturally corresponds to a given piece of advice:
Advice = "Don't touch any mice unless you have gloves on."
Grounding: Effect
Advice = "Walking into lava will kill you."
Grounding: Effect
Advice = "First get the money, then go to the green square."
Grounding: Plan
Advice = "Go through the door to the goal."
Grounding: Plan
Advice = "If you have the key, go to the door, otherwise you need to get the key."
Grounding: Policy
Advice = "If there are any closed doors, open them."
Grounding: Policy
Advice = "Open any doors if they are closed."
Grounding: Policy

A.3.2 [VIRTUALHOME] PROMPT USED FOR STAGE 2 OF THE PIPELINE TO TRANSLATE A PIECE OF ADVICE INTO AN RLANG PLAN.

Your task is to translate natural language advice to RLang plan, which is a sequence of specific steps to take. For each instance, we provide a piece of advice in natural language, a list of allowed primitives, and you should complete the instance by filling the missing plan function. Don't use any primitive outside the provided primitive list corresponding to each instance, e.g., if there is no 'green_door' in the primitive list you must not use 'green_door' for the plan function.

Advice = "Open the door with the key and go through it to the goal"
Primitives = ['Agent', 'Wall', 'GoalTile', 'Lava', 'Key', 'Door', 'Box', 'Ball', 'left', 'right', 'forward', 'pickup', 'drop', 'toggle', 'done', 'pointing_right', 'pointing_down', 'pointing_left', 'pointing_up', 'go_to', 'step_towards', 'yellow_key', 'yellow_door', 'agent', 'goal', 'at', 'in_inventory']

```
Plan main:
  Execute go_to(yellow_key)
  Execute pickup
  Execute go_to(yellow_door)
  Execute toggle
  Execute go_to(goal)
```

Advice = "Get the key behind the red door to open the grey door. Then drop the key to the left."
Primitives = ['Agent', 'Wall', 'GoalTile', 'Lava', 'Key', 'Door', 'Box', 'Ball', 'left', 'right', 'forward', 'pickup', 'drop', 'toggle', 'done', 'pointing_right', 'pointing_down', 'pointing_left', 'pointing_up', 'go_to', 'step_towards', 'yellow_key', 'yellow_door', 'agent', 'goal', 'at', 'in_inventory']

```
Plan main:
  Execute go_to(red_door)
  Execute toggle
  Execute go_to(grey_key)
  Execute pickup
  Execute go_to(grey_door)
  Execute toggle
  Execute left
  Execute drop
```

Advice = "Get the key behind the red door to open the grey door." Primitives = ['Agent', 'Wall', 'GoalTile', 'Lava', 'Key', 'Door', 'Box', 'Ball', 'left', 'right', 'forward', 'walk_to', 'open', 'close', 'putin', 'grab', 'inside', 'grey_key_11', 'red_door', 'grey_door_127', 'agent', 'purple_ball', 'is_on_a', 'at', 'at_any', 'in_inventory']

```
Plan main:
  Execute walk_to(red_door)
  Execute open(red_door)
  Execute walk_to(grey_key_11)
  Execute grab(grey_key_11)
  Execute walk_to(grey_door_127)
  Execute open(grey_door_127)
```

1188
1189
1190
1191
1192
1193
1194
1195
1196
1197
1198
1199
1200
1201
1202
1203
1204
1205
1206
1207
1208
1209
1210
1211
1212
1213
1214
1215
1216
1217
1218
1219
1220
1221
1222
1223
1224
1225
1226
1227
1228
1229
1230
1231
1232
1233
1234
1235
1236
1237
1238
1239
1240
1241

### A.3.3 [VIRTUALHOME] PROMPT USED FOR STAGE 2 OF THE PIPELINE TO TRANSLATE A PIECE OF ADVICE INTO AN RLANG POLICY.

Your task is to translate natural language advice to RLang policy, which is a direct function from states to actions. For each instance, we provide a piece of advice in natural language, a list of allowed primitives, and you should complete the instance by filling the missing policy function. Don't use any primitive outside the provided primitive list corresponding to each instance, e.g., if there is no 'green_door' in the primitive list you must not use "green_door' for the policy function.

Advice = "If the yellow door is open, go through it and walk to the goal. Otherwise open the yellow door if you have the key."
Primitives = ['Agent', 'Wall', 'GoalTile', 'Lava', 'Key', 'Door', 'Box', 'Ball', 'left', 'right', 'forward', 'pickup', 'drop', 'toggle', 'done', 'pointing_right', 'pointing_down', 'pointing_left', 'pointing_up', 'go_to', 'step_towards', 'yellow_key', 'yellow_door', 'agent', 'goal', 'at', 'carrying']

```
Policy main:
  if yellow_door.is_open:
    Execute go_to(goal)
  elif carrying(yellow_key) and at(yellow_door) and not yellow_door.is_open:
    Execute toggle
```

Advice = "If you don't have the key, go get it"
Primitives = ['Agent', 'Wall', 'GoalTile', 'Lava', 'Key', 'Door', 'Box', 'Ball', 'left', 'right', 'forward', 'pickup', 'drop', 'toggle', 'done', 'pointing_right', 'pointing_down', 'pointing_left', 'pointing_up', 'go_to', 'step_towards', 'grey_key_11', 'red_door', 'grey_door', 'agent', 'purple_ball', 'is_on_a', 'at', 'at_any', 'in_inventory']

```
Policy main:
  if at(grey_key_11):
    Execute pickup
  elif not carrying(grey_key_11):
    Execute go_to(grey_key_11)
```

Advice = "If you're at the fridge, close it."
Primitives = ['Toothpaste', 'Bedroom', 'Character', 'Cereal', 'Bathroom', 'Sofa', 'Cabinet', 'Salmon', 'Pie', 'Kitchentable', 'Remotecontrol', 'Fridge', 'Microwave', 'Kitchen', 'Bookshelf', 'Livingroom', 'walk_to', 'open', 'close', 'putin', 'puton', 'grab', 'drop', 'can_drop', 'is_drop', 'inside', 'inside_something', 'on', 'at', 'is_closed', 'is_open', 'holding', 'near', 'character_1', 'kitchen_205', 'bookshelf_249', 'fridge_305', 'oven_133', 'pie_319', 'chicken_127', 'cabinet_19']

```
Policy main:
  if at(fridge_305):
    Execute close(fridge_305)
```

1242
1243
1244
1245
1246
1247
1248
1249
1250
1251
1252
1253
1254
1255
1256
1257
1258
1259
1260
1261
1262
1263
1264
1265
1266
1267
1268
1269
1270
1271
1272
1273
1274
1275
1276
1277
1278
1279
1280
1281
1282
1283
1284
1285
1286
1287
1288
1289
1290
1291
1292
1293
1294
1295

### A.3.4 [VIRTUALHOME] PROMPT USED FOR STAGE 2 OF THE PIPELINE TO TRANSLATE A PIECE OF ADVICE INTO AN RLANG EFFECT.

Your task is to translate natural language advice to RLang effect, which is a prediction about the state of the world or the reward function. For each instance, we provide a piece of advice in natural language, a list of allowed primitives, and you should complete the instance by filling the missing effect function. Don't use any primitive outside the provided primitive list corresponding to each instance, e.g., if there is no 'green_door' in the primitive list you must not use 'green_door' for the effect function.

Advice = "Don't go to the door without the key"
Primitives = ['yellow_door', 'goal', 'pickup', 'yellow_key', 'toggle', 'go_to', 'carrying', 'at']

```
Effect main:
  if at(yellow_door) and not carrying(yellow_key):
    Reward -1
```

Advice = "Don't walk into closed doors, since it takes no effect"
Primitives = ['Agent', 'Wall', 'GoalTile', 'Lava', 'Key', 'Door', 'Box', 'Ball', 'left', 'right', 'forward', 'pickup', 'drop', 'toggle', 'done', 'pointing_right', 'pointing_down', 'pointing_left', 'pointing_up', 'go_to', 'step_towards', 'agent', 'goal', 'is_on_a', 'at', 'at_any', 'in_inventory']

```
Effect main:
  if at(yellow_door) and not yellow_door.is_open and A == forward:
    Reward -1
    S' -> S
```

Advice = "Walking to a broken object won't do anything. You can't grab the ball if it's inside something."
Primitives = ['Agent', 'Wall', 'GoalTile', 'Lava', 'Key', 'Door', 'Box', 'Ball', 'is_broken', 'left', 'right', 'forward', 'grab', 'drop', 'toggle', 'done', 'pointing_right', 'pointing_down', 'pointing_left', 'pointing_up', 'inside_something', 'go_to', 'step_towards', 'agent', 'goal', 'is_on_a', 'at', 'at_any', 'in_inventory', 'gate_12', 'door_16', 'ball_121']

```
Effect main:
  if A == walk_to(gate_12) and is_broken(gate_12):
    S' -> S
  if A == walk_to(door_16) and is_broken(door_16):
    S' -> S
  if A == grab(ball_121) and inside_something(ball_121):
    S' -> S
```

Advice = "Don't go to the purple ball"
Primitives = ['Agent', 'Wall', 'GoalTile', 'Lava', 'Key', 'Door', 'Box', 'Ball', 'left', 'right', 'forward', 'walk_to', 'open', 'close', 'putin', 'grab', 'inside', 'holding', 'grey_key_11', 'red_door', 'grey_door_127', 'agent', 'purple_ball', 'is_on_a', 'at', 'at_any', 'in_inventory']

```
Effect main:
  if A == walk_to(purple_ball):
    Reward -1
```

Advice = "If you put the pie into the microwave and the chicken into the oven, and make sure that they are both on, you will get reward and the episode will end."

Primitives = ['Toothpaste', 'Bedroom', 'Character', 'Cereal', 'Bathroom', 'Sofa', 'Cabinet', 'Salmon', 'Pie', 'Kitchentable', 'Remotecontrol', 'Fridge', 'Microwave', 'Kitchen', 'Bookshelf', 'Livingroom', 'walk_to', 'open', 'turn_on', 'close', 'putin', 'puton', 'grab', 'drop', 'can_drop', 'is_drop', 'inside', 'inside_something', 'on', 'at', 'is_closed', 'is_open', 'holding', 'near', 'character_1', 'kitchen_205', 'bookshelf_249', 'fridge_305', 'oven_133', 'pie_319', 'chicken_127', 'microwave_19']

```
Effect main:
  if inside(pie_319, microwave_19) and inside(chicken_127, oven_133):
    if is_closed(microwave_19) and at(oven_133) and A == turn_on(oven_133):
      Reward 5
      S' -> S
    elif is_closed(oven_133) and at(microwave_19) and A == turn_on(microwave_19):
      Reward 5
      S' -> S
```

Advice = "If you're not trying to pick up the fridge, you will be penalized" Primitives = ['Sofa', 'Kitchentable', 'Bathroom', 'Salmon', 'Kitchen', 'Bookshelf', 'Cereal', 'Cabinet', 'Livingroom', 'Fridge', 'Bedroom', 'Character', 'Toothpaste', 'Pie', 'Microwave', 'Remotecontrol', 'walk_to', 'open', 'close', 'putin', 'puton', 'grab', 'drop', 'can_drop', 'is_drop', 'inside', 'inside_something', 'on', 'at', 'fridge_305', 'is_pickup', 'is_closed', 'is_open', 'holding', 'near', 'character_1', 'bathroom_11', 'toothpaste_62', 'bedroom_73', 'kitchen_205', 'kitchentable_231', 'bookshelf_249', 'fridge_305', 'microwave_313', 'pie_319', 'salmon_327', 'cereal_334', 'livingroom_335', 'sofa_368', 'cabinet_415', 'remotecontrol_452']

```
Effect main:
  if fridge_305(fridge_305) and not is_pickup(A):
    Reward -1
```

Advice = "if you have any key for a door that you cannot reach, you should drop it"

Primitives = ['Agent', 'Wall', 'GoalTile', 'Lava', 'Key', 'Door', 'Box', 'Ball', 'left', 'right', 'forward', 'pickup', 'drop', 'toggle', 'done', 'pointing_right', 'pointing_down', 'pointing_left', 'pointing_up', 'go_to', 'step_towards', 'green_ball', 'green_box', 'purple_box', 'agent', 'purple_ball', 'at', 'reachable', 'carrying']

```
Policy main:
  if carrying(green_key) and not reachable(green_door):
    Execute drop
  if carrying(purple_key) and not reachable(purple_door):
    Execute drop
  if carrying(red_key) and not reachable(red_door):
    Execute drop
```

Advice = "Hey listen, you can open the door if you have the key and at the door when the door is closed"

Primitives = ['Agent', 'Wall', 'GoalTile', 'Lava', 'Key', 'Door', 'Box', 'Ball', 'left', 'right', 'forward', 'pickup', 'drop', 'toggle', 'done', 'pointing_right', 'pointing_down', 'pointing_left', 'pointing_up', 'go_to', 'step_towards', 'green_ball', 'green_box', 'purple_box', 'agent', 'purple_ball', 'at', 'reachable', 'carrying']

```
Policy main:
  if carrying(purple_key) and not purple_door.is_open and at(purple_door):
    Execute toggle
```

## A.4 USER STUDY - TRANSLATED ADVICE FOR MINIGRID EXPERIMENTS

Table 3: Advice from the user study translated to RLang.

| Language Advice | RLang Translation |
|---|---|
| "Remember to toggle to open doors." | ```Policy main:
  if at(yellow_door) and not yellow_door.is_open
  :
    Execute toggle
  elif at(red_door) and not red_door.is_open:
    Execute toggle
  elif at(purple_door) and not purple_door.
  is_open:
    Execute toggle
  elif at(blue_door) and not blue_door.is_open:
    Execute toggle
  elif at(green_door) and not green_door.is_open
  :
    Execute toggle
  elif at(grey_door) and not grey_door.is_open:
    Execute toggle``` |
| "You don't need to carry keys to open the grey door." | ```Effect main:
  if at(grey_door) and carrying(red_key):
    S' -> S
    Reward -1
  elif at(grey_door) and carrying_something():
    S' -> S
    Reward -1``` |
| "Identify the room with the red key, move to that room by opening the door. Pick up the key. Identify the room with the red door, proceed there. Open the red door. Find the green square and go there to finish the game." | ```Plan main:
  Execute go_to(red_key)
  Execute pickup
  Execute go_to(red_door)
  Execute toggle
  Execute go_to(goal)``` |
| "Move to the grey door, open it and enter the room until you get to the red key, pick it up. Exit the room and move towards the red door, open it and get into that room. Move to the green block and enter it." | ```Plan main:
  Execute go_to(grey_door)
  Execute toggle
  Execute go_to(red_key)
  Execute pickup
  Execute go_to(grey_door)
  Execute toggle
  Execute go_to(red_door)
  Execute toggle
  Execute go_to(goal)``` |
| "Go to the grey door. open the grey door. go to the red key. pick up the red key. go to the red door. open the red door. go to the green square." | ```Plan main:
  Execute go_to(grey_door)
  Execute toggle
  Execute go_to(red_key)
  Execute pickup
  Execute go_to(red_door)
  Execute toggle
  Execute go_to(goal)``` |

1404
1405
1406
1407
1408
1409
1410
1411
1412
1413
1414
1415
1416
1417
1418
1419
1420
1421
1422
1423
1424
1425
1426
1427
1428
1429
1430
1431
1432
1433
1434
1435
1436
1437
1438
1439
1440
1441
1442
1443
1444
1445
1446
1447
1448
1449
1450
1451
1452
1453
1454
1455
1456
1457

Table 4: Advice from the user study translated to RLang (continued).

| Language Advice | RLang Translation |
|---|---|
| "Pick up the red key after opening the grey door. Then walk to the red door, open it, and go to the goal." | ```Plan main:   Execute go_to(grey_door)   Execute toggle   Execute go_to(red_key)   Execute pickup   Execute go_to(red_door)   Execute toggle   Execute go_to(goal)``` |
| "You cannot open the red door without a red key." | ```Effect main:   if at(red_door) and not carrying(red_key):     S' -> S     Reward -1``` |
| "Walking towards the red door is not very useful if it is closed." | ```Effect main:   if at(red_door) and not(red_door.is_open) and   A == forward:     S' -> S     Reward -1``` |
| "Go down until the second door on the left and pick up the key. Then exit the room and go down until the next door on the left and use it to open the door and get to the green box." | ```Plan main:   Execute go_to(second_left_door)   Execute pickup   Execute go_to(exit)   Execute go_to(next_left_door)   Execute toggle   Execute go_to(green_box)``` |
| "Go to the room that has the red key, pick it up, and then go to the room with a red door. Enter the room, and go to the green goal object." | ```Plan main:   Execute go_to(red_key)   Execute pickup   Execute go_to(red_door)   Execute toggle   Execute go_to(goal)``` |

## A.5 MIDMAZELAVA - TRANSLATED ADVICE

Advice: "Pick up the blue ball and drop it to your right. Then pick up the green key and unlock the green door. Then drop the key to your right. Some general advice: If you are carrying a key and its corresponding door is closed, open the door if you are at it, otherwise go to the door if you can reach it. Otherwise, drop any keys for doors you can't reach. If you can reach the goal, go to it. Walking into lava will kill you. If you're not at a door, toggling will do nothing. Trying to pick something up while you're carrying something is pointless. Walking into walls will do nothing."

```
Plan main:
  Execute go_to(blue_ball)
  Execute pickup
  Execute right
  Execute drop
  Execute go_to(green_key)
  Execute pickup
  Execute go_to(green_door)
  Execute toggle
  Execute right
  Execute drop

Policy main:
  if carrying(green_key) and not green_door.is_open:
    if at(green_door):
      Execute toggle
    elif reachable(green_door):
      Execute go_to(green_door)

  elif carrying(grey_key) and not grey_door.is_open:
    if at(grey_door):
      Execute toggle
    elif reachable(grey_door):
      Execute go_to(grey_door)

  elif reachable(goal):
    Execute go_to(goal)

  elif carrying(green_key) and not reachable(green_door):
    Execute drop

  elif carrying(grey_key) and not reachable(grey_door):
    Execute drop

Effect main:
  if at(Lava) and A == forward:
    S' -> S*0
    Reward -1
  if not at(Door) and A == toggle:
    S' -> S
    Reward 0
  if carrying_something() and A == pickup:
    S' -> S
    Reward 0
  if at(Wall) and A == forward:
    S' -> S
    Reward 0
```

## A.6 HARDMAZELIGHT - TRANSLATED ADVICE

Advice: "Go and pick up the green ball, and drop it on your left, and then go pick up the blue key, and go to the blue door and open it up and drop the key on your left, and then go pick up the green key, and go to the green door to open it and drop the key on your left, and then go pick up the purple ball and drop it on your right. Nothing will happen if you walk towards the wall, or try to open a purple door without the purple key if it is locked. The applies for the yellow door and key as well as the red door and key. If you can reach the grey door and it is closed but you have the key, open it if you are at it or otherwise go to it. The same applies to the purple door, yellow door, and red door. Lastly, if you find the goal is reachable just go to the goal directly."

```
Plan main:
  Execute go_to(green_ball)
  Execute pickup
  Execute left
  Execute drop
  Execute go_to(blue_key)
  Execute pickup
  Execute go_to(blue_door)
  Execute toggle
  Execute left
  Execute drop
  Execute go_to(green_key)
  Execute pickup
  Execute go_to(green_door)
  Execute toggle
  Execute right
  Execute drop
  Execute go_to(purple_ball)
  Execute pickup
  Execute right
  Execute drop

Effect main:
  if at(Wall) and A == forward:
    Reward 0
    S' -> S
  elif at(purple_door) and purple_door.is_locked and A == toggle and not carrying(
  purple_key):
    Reward 0
    S' -> S
  elif at(yellow_door) and yellow_door.is_locked and A == toggle and not carrying(
  yellow_key):
    Reward 0
    S' -> S
  elif at(red_door) and red_door.is_locked and A == toggle and not carrying(red_key):
    Reward 0
    S' -> S
```

```
Policy main:
  if reachable(grey_door) and carrying(grey_key) and grey_door.is_locked:
    if at(grey_door):
      Execute toggle
    else:
      Execute go_to(grey_door)

  elif reachable(purple_door) and carrying(purple_key) and purple_door.is_locked:
    if at(purple_door):
      Execute toggle
    else:
      Execute go_to(purple_door)

  elif reachable(yellow_door) and carrying(yellow_key) and yellow_door.is_locked:
    if at(yellow_door):
      Execute toggle
    else:
      Execute go_to(yellow_door)

  elif reachable(red_door) and carrying(red_key) and red_door.is_locked:
    if at(red_door):
      Execute toggle
    else:
      Execute go_to(red_door)

  elif reachable(goal):
    Execute go_to(goal)
```

## A.7 FOODSAFETY - TRANSLATED ADVICE

Advice: "Go to fridge and open it, and then go find the pie and pick it up, walk back to the fridge and put the pie in the fridge. You have to close the fridge too", "If the salmon is in the microwave, and you are at the microwave and it's open, close it. Otherwise if you are holding salmon, do the following: open the microwave if you are near it but it's closed, put the salmon into the microwave if it's open and you're near it, else walk to the microwave.", "If the pie is in the fridge, and the salmon is in the microwave, then closing the fridge if the microwave is closed or closing the microwave if the fridge is closed will give you reward and end the episode."

```
Plan main:
  Execute walk_to(fridge_305)
  Execute open(fridge_305)
  Execute walk_to(pie_319)
  Execute grab(pie_319)
  Execute walk_to(fridge_305)
  Execute putin(fridge_305)
  Execute close(fridge_305)

Policy main:
  if inside(salmon_327, microwave_313) and at(microwave_313) and is_open(microwave_313):
    Execute close(microwave_313)
  elif holding(salmon_327):
    if at(microwave_313) and is_closed(microwave_313):
      Execute open(microwave_313)
    elif at(microwave_313) and is_open(microwave_313):
      Execute putin(microwave_313)
    else:
      Execute walk_to(microwave_313)

Effect main:
  if inside(pie_319, fridge_305) and inside(salmon_327, microwave_313):
    if is_closed(fridge_305) and at(microwave_313) and A == close(microwave_313):
      Reward 5
      S' -> S
    elif is_closed(microwave_313) and at(fridge_305) and A == close(fridge_305):
      Reward 5
      S' -> S
```

## A.8   COUCHPOTATO - TRANSLATED ADVICE

Advice: "If you're holding the toothpaste and can drop it, drop it.", "Go grab the remote control and put it on the sofa.", "If you're holding the toothpaste and not trying to drop it, you will be penalized. Also, nothing will happen if you try to walk to the remote control, cereal, toothpaste, or salmon, if you try to walk to them and they are contained inside anything."

```
Effect main:
  if holding(toothpaste_62) and not is_drop(A):
    Reward -1
  if inside_something(remotecontrol_452) and A == walk_to(remotecontrol_452):
    S' -> S
  if inside_something(cereal_334) and A == walk_to(cereal_334):
    S' -> S
  if inside_something(toothpaste_62) and A == walk_to(toothpaste_62):
    S' -> S
  if inside_something(salmon_327) and A == walk_to(salmon_327):
    S' -> S

Policy main:
  if holding(toothpaste_62) and can_drop(toothpaste_62):
    Execute drop(toothpaste_62)

Plan main:
  Execute walk_to(remotecontrol_452)
  Execute grab(remotecontrol_452)
  Execute walk_to(sofa_368)
  Execute puton(remotecontrol_452, sofa_368)
```

A.9  GROUNDING SEMANTICALLY-AMBIGUOUS ADVICE

> Advice: Put the baked good into the white box.
> 'the baked good' refers to Pie, and 'the white box' refers to Microwave.
> Advice: Put the baked dessert in the red food-box.
> 'the baked dessert' refers to Pie, and 'the red food-box' refers to Fridge.
> Advice: Store the pastry in the tall red box.
> 'the pastry' refers to Pie, and 'the tall red box' refers to Fridge.
> Advice: Open the appliance for heating.
> 'the appliance for heating' refers to Microwave.
> Advice: Open the cooking device.
> 'the cooking device' refers to Microwave.
> Advice: Grab the blue box and put it on the wooden white surface.
> In this context, 'the blue box' refers to Cereal, and 'the wooden white surface' refers to Kitchentable.
> Advice: Place the seafood on the dining surface.
> 'the seafood' refers to Salmon and 'the dining surface' refers to Kitchentable.
> Advice: Store the breakfast grains on the black shelving.
> 'the breakfast grains' refers to Cereal, and 'the black shelving' refers to Bookshelf.
> Advice: Set the channel changer on the seating furniture.
> 'the channel changer' refers to Remotecontrol, and 'the seating furniture' refers to Sofa.
> Advice: Open the cooling appliance and grab the seafood.
> 'the cooling appliance' refers to Fridge, and 'the seafood' refers to Salmon.
> Advice: Put the small tube on the wooden box.
> 'the small tube' refers to Toothpaste, and 'the wooden box' likely refers to Cabinet.
> Advice: Put the food with the white packaging into the big red box
> 'the food with the white packaging' refers to Cereal, and 'the big red box' refers to Fridge.
> Advice: Place the baked dessert on the white hard surface.
> 'the baked dessert' refers to Pie, and 'the white hard surface' refers to Kitchentable.
> Advice: Place the seafood on the white wooden surface.
> 'the seafood' refers to Salmon, and 'the white wooden surface' refers to Kitchentable.
> Advice: Put the orange food in the red box.
> 'the orange food' refers to Salmon, and 'the red box' refers to Fridge.
> Advice: Open the red food-box.
> 'the red food-box' refers to Fridge.
> Advice: Open the white box you might put food in.
> 'the white box you might put food in' refers to Microwave.

B  ADDITIONAL EXPERIMENT: GROUNDING COMMANDS TO RLANG PLANS - COMPARISON TO SAYCAN

We compare our method to SayCan (Ahn et al., 2022), which uses the commonsense reasoning capacity of LLMs to satisfy a natural language request by generating a simple plan consisting of a series of pre-engineered high-level robot skills. Adopting the same

In this experiment we demonstrate that, in a simulated 3-dimensional physical environment, RLang can express the full range of natural language instructions necessary for a robot to complete various tasks. By grounding natural language instructions to RLang policies over this environment, we achieve performance on par with the results from the open-source tasks that the original SayCan paper evaluated on, showing that RLang can be easily substituted for the formal language that the SayCan authors developed for this specific task, allowing for generalization without sacrificing performance.

Similar to the SayCan work, we assume that we are given a grounding tuple $\langle \Pi, S, A, \rangle$, and a set of skills $\Pi$, where each skill $\pi \in \Pi$ performs an action with the robot arm to manipulate a block or a bowl. We evaluate on the 8 unique tasks made available in the open-source version of SayCan, running each task across 10 different randomly selected initial states, using both the native SayCan language and RLang as the DSL for grounding natural language instructions to robot behavior.

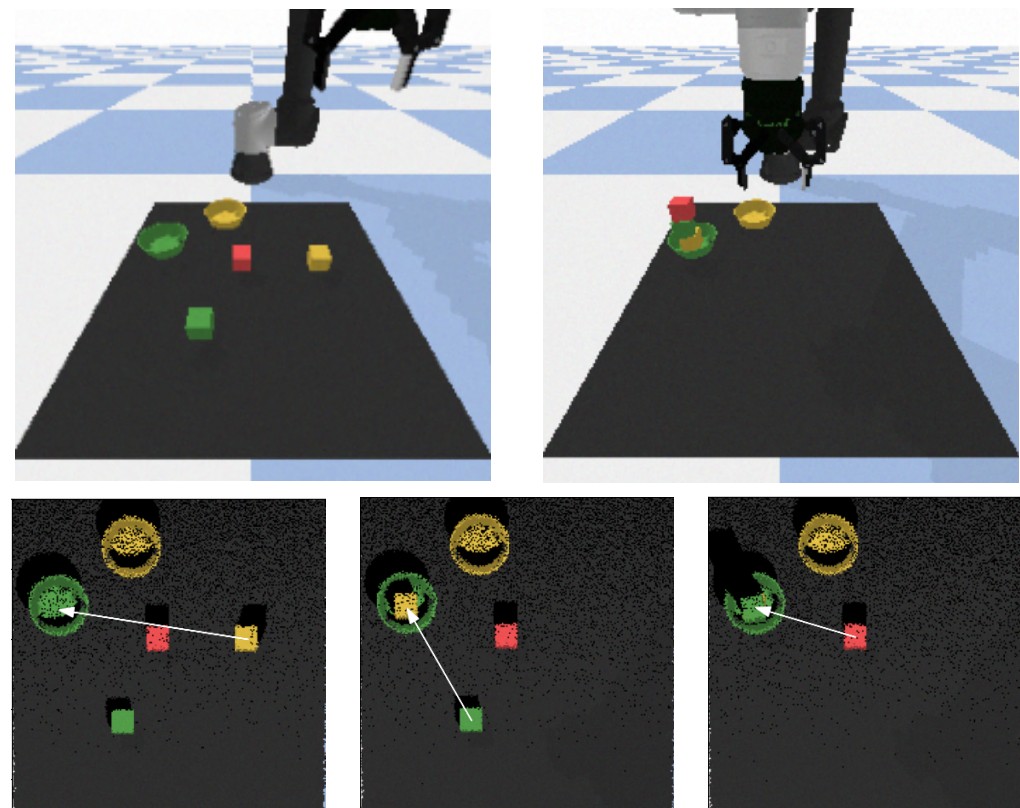

Figure 10: Top: One configuration of the initial and completed states in the SayCan environment. Bottom: the action sequence to execute on the instruction: "put all the blocks in the green bowl.", from the robot arm's perspective

Table 5: Success rates of SayCan and RLang-based instruction grounding rate on each task, out of 10 random initial states.

| Instruction | SayCan | NL2RLang |
|---|---|---|
| put all the blocks in different corners. | 10 | 10 |
| move the block to the bowl. | 6 | 6 |
| put any blocks on their matched colored bowls. | 7 | 7 |
| put all the blocks in the green bowl. | 7 | 7 |
| stack all the blocks. | 8 | 8 |
| make the highest block stack. | 7 | 7 |
| put the block in all the corners. | 10 | 10 |
| clockwise, move the block through all the corners. | 10 | 10 |

Each task configuration that the original SayCan agent completes, is also completed by the RLang agent. While their behavior on failure cases occasionally varied, these were generally caused by errors in the vision model's processing of shadows in the simulated environment. These generally caused the textual scene description fed into GPT-3 to include a block where a bowl should be, and occasionally incorrect color labels, which often provided the text-only planner with a nonsensical task that was impossible to complete. Similarly, in cases where multiple action orders could satisfy the request, the RLang and SayCan pipelines occasionally diverged in the order of actions. Nonetheless, neither language grounding pipeline completed a task configuration that the other one did not.

## C  EXPERIMENT PARAMETERS

In all experiments, for both Dyna-Q and RLang-Dyna-Q, we set the learning rate $\alpha$ to 0.1, the discount factor $\gamma$ to 0.99, $\epsilon_1 = \epsilon_2 = 0.1$, except when there is policy or plan advice, uniform exploration, and 16 hallucinatory updates with the learned dynamics model.

For the translation step, we use the `gpt-3.5-turbo-instruct` model with a temperature of 0.

