# OpenReview forum: "Informing Reinforcement Learning Agents by Grounding Language to Markov Decision Processes"
_ICLR.cc/2025/Conference — Submitted to ICLR 2025_

### Official Review · Reviewer_8geU · 2024-10-25

**Soundness:** 3
**Presentation:** 3
**Contribution:** 2
**Rating:** 6
**Confidence:** 3

**Summary:**

The paper proposed RLang-Dyna-Q, which can ground any language advice to all components in MDP, compared to grounding only to individual components before. The solution uses in-context learning to first select the grounding type, then translate the advice to RLang program. The enhancement outperforms prior approaches.

**Strengths:**

1. The algorithm automates language-to-MDP component translation, and streamlines the process of learning human decision-making for robots
2. The authors conducted extensive experiments and described the algorithm clearly

**Weaknesses:**

1. In-context learning limits the capability enhancement of the language model. It might be better if we could make the LM trainable and train the language model and the RL system end-to-end
2. Human language might not be expressive enough to be translated to RLang. In the experiment section, it stated that some advice cannot be converted to RLang. Could we have a more natural intermediate representation for the advice and agent?

**Questions:**

1. How to go beyond in-context learning?
2. How to handle the inexpressiveness of human language for RLang?

---

> ### Author Response · Authors · 2024-11-13
>
> Thank you for your insightful observations and questions. We've responded to them below.
>
> Regarding Weakness 1 and Question 1:
>
> This is a good observation, a more integrated system might train a language model and an RL agent end-to-end. For the goals of this paper, however, in-context translation sufficed to show that language could be used to inform RL agents. We acknowledge this interesting research direction and would be excited to see it addressed in future work.
>
> Regarding Weakness 2 and Question 2:
>
> The reviewer raises an important question about the expressivity of natural language and RLang, and how variations in expressivity between the two languages can make effective translation difficult. These concepts have been explored somewhat in the machine translation and semantic parsing communities. A potential solution—as the reviewer suggests—might be to introduce another intermediate representation for natural language such as LambdaDCS, a general logical form, and compile this language into RLang, an MDP-theoretic specification language. We note, however, that RLang enables the introduction of novel syntax via Vocabulary Files, which we have leveraged in these experiments to increase the expressivity of RLang itself, bypassing the need for another intermediate language. In future work, we hope to automate this process of semantic expansion so that more language may be grounded using this methodology.
>
> We welcome any additional questions or critiques.

---

> > ### Author Response · Authors · 2024-11-21
> >
> > We appreciate the time you have taken to provide feedback on our submission. We have worked diligently to address your comments and concerns. If our responses and updates resolve your questions, we kindly ask you to consider revisiting your score. Please do not hesitate to let us know if you have additional feedback or concerns.

---

> > > ### Comment · Reviewer_8geU · 2024-11-25
> > > **Thanks for the response and i will keep the original score**
> > >
> > > Agree with weakness 1. In-context learning is more cost efficient and more accessible than finetuning.
> > > For weakness 2, looking forward to the new technique that can resolve the inexpressiveness because this is an essential part for the practical adoption.

---

### Official Review · Reviewer_3iuj · 2024-10-27

**Soundness:** 3
**Presentation:** 3
**Contribution:** 3
**Rating:** 6
**Confidence:** 3

**Summary:**

This paper studies the problem of leveraging natural language advice for training effective/efficient RL agents. The authors hypothesize that some natural language advice is more suitable to be grounded in reward function while others are better captured in transition functions. They further suggest that leveraging natural language advice by grounding them in a formal language (RLang) that is capable of describing all aspects of the system, is better than advising a subset of the system.

The authors adapt Dyna-Q to work with grounded advice in RLang. They evaluate this method in Minigrid and VirtualHome with mostly positive results to support their hypothesis.

**Strengths:**

1. I appreciate the range of experiments included, as well as a comparison with SayCan in the appendix.
2. I also enjoy reading the section on user studies, and section 4.3 on automated semantic labeling for disambiguation advice. In general, I agree that gradually removing the need for expert annotations is important, let it be dense natural language advice, crafted vocabulary, or RLang grounding files in this case.
3. This is an important research topic and the contribution is contextualized nicely.
4. Most of the paper is quite clear. A few improvements can be made - see weakness 3.

**Weaknesses:**

1. Expert advice seems much more dense, verbose, and low-level (almost program-like) than non-expert advice. It is not completely surprising to me that LLMs can ground them to predefined symbols that are approximately defined on a similar level of abstraction.
2. It might help to have a paragraph discussing results and how advice on effect/policy/plan each contributes to the combined policy. Are they the same? Is it task-dependent? I think this can help better justify that an approach to encode "information about every element of an MDP" is necessary.

(The two concerns above are why I gave 2 for contribution and not higher. Would be happy to improve scores if they are addressed)

3. Stylistic nit-picking: could you please increase the font size in Figure 1, and reward curves in Figure 2-6? The text in Figure 7 looks much better. Perhaps capitalize "Perception" in the title in the left figure for consistency. Consistent legend colors and orders for different methods on page 8 would improve comparability across figures.
4. Broken reference on line 164.

**Questions:**

1. How was the expert advice (textboxes on page 8) collected for the main experiments (who are the experts, how are they trained, what's the protocol)?
2. Do you have ablation studies for HardMaze in Figure 4?
3. Why is RLang-Dyna-Q-combined worse than RLang-Dyna-Q-plan curve in Figure 6?

---

> ### Author Response · Authors · 2024-11-13
>
> We sincerely thank the reviewer for their valuable feedback and thoughtful comments. We address their comments below.
>
> **Weaknesses**
>
> Regarding Weakness 1:
>
> Your point that the dense, verbose, and low-level expert advice is potentially at a similar level of abstraction to the pre-defined symbols is well-taken. However, we don’t suggest that this should be surprising, and argue that more granular advice is objectively easier to operationalize than more abstract advice. Such advice relies less on the common-sense capabilities of LLMs and more on their capacity to perform approximate machine translation.
>
> Regarding Weakness 2:
>
> We agree with your suggestion to elucidate how various kinds of advice impact performance, and have included the following paragraph at the end of section 4.2 to address these questions. Unfortunately this will displace the table of results for the user study to the appendix, but we believe this analysis is more central to the thrust of the paper.
>
> > The impact of each kind of advice (e.g. plans, policies, transitions, and rewards) varied across tasks in the VirtualHome and Minigrid experiments, with some environments benefiting primarily from plan-centric advice and others benefiting most from policy advice. In virtually all cases, model-centric advice---about transitions and rewards---was less valuable than other forms of advice. We suggest that this discrepancy is due to how useful model-based advice is in comparison to explicit policy and planning advice. While policy and planning advice describe which actions to take in a given context, model-based advice was often used to suggest which actions \textit{not} to take, relying on the underlying learning agent to find the best action. Furthermore, model-based advice was useful less of the time, i.e. in fewer states. This is best illustrated by comparing the relative performance of effect-enabled RLang-Dyna-Q agents with policy and plan-enabled agents in the MidMazeLava Experiment in Figure \ref{fig:midmazelava} and the FoodSafety Experiment in Figure \ref{fig:foodsafety}. The model-based advice in the first experiment is to avoid lava, which there are many opportunities to walk into, resulting in the performance of the effect-enabled agent closer to the plan and policy-enabled agents. By comparison, the model-based advice in the second experiment is more niche, accounting only for a handful of transitions, and the effect-enabled agent correspondingly performs closer to baseline Dyna-Q than to the plan and policy-enabled agents.
> >
>
> Regarding Weaknesses 3 and 4:
>
> We have increased the fonts in those figures from size 14 to 17 for smaller text and from 20 to 24 for titles and axes labels. We hope they are more legible now! We have also made the text casing match in Figure 7.
>
> **Questions**
>
> Regarding Question 1:
>
> Expert advice was given by two students who were familiar with the environments, including the action (i.e. a go-to skill) and perception space of the agent (e.g. that the agent sees the world in terms of objects and primitive predicates).
>
> We added an aside to a sentence in the third paragraph in section 4.1 explaining a bit about human experts:
>
> > For each environment, we collected multiple pieces of natural language advice from human experts---people familiar with both the environment and how the agent interacts with it via perception and action, i.e. the skills the agent has access to and the fact that its perception consists of objects and simple predicates
> >
>
> Regarding Question 2:
>
> We do not have ablations for HardMaze. In contrast with the other MiniGrid environments, DynaQ was not able to achieve any reward in this environment due to its long-horizon nature. Our goal with this experiment was to show that language advice could make the difference between some reward and none at all — as you may notice, the returns were relatively modest compared to other environments, but significant.
>
> Regarding Question 3:
>
> The combined agent is worse than the plan-informed agent due to the effect advice, which decreases performance due to non-determinism in the VirtualHome simulator which is an unintended bug and not a feature of the environment. We note this in the description for Figure 6.
>
> We again thank the reviewer and welcome any additional questions, comments, and concerns they may have.

---

> > ### Comment · Reviewer_3iuj · 2024-11-14
> >
> > I appreciate the authors' timely response. They have adequately addressed my comments about weaknesses 3-4 and my questions.
> >
> > Weakness 1 (or rather a limitation): In my opinion, assuming access to detailed, almost program-like natural language advice sounds not so different from directly defining the RLang files (perhaps with Copilot in a comment-tab style). This is not a detrimental weakness but still a limitation and could be a bit disappointing for folks looking for ways to handle more natural advice like the non-expert advice in this work.
> >
> > Weakness 2: I appreciate the analysis added by the authors. Overall, I am not sure if the experimental evidence (at least in the environments presented) is strong enough to support the argument that grounding every aspect in an MDP is necessary. The demonstrated *feasibility* of grounding every part is very impressive though.
> >
> > For these reasons, I will raise the contribution score from 2 to 3, and maintaining the overall score.

---

> > > ### Author Response · Authors · 2024-11-21
> > >
> > > We thank the reviewer for their consideration -- we generally agree with your points, and we believe the greater community would be interested in the *feasibility* of this approach.

---

### Official Review · Reviewer_EByn · 2024-10-28

**Soundness:** 3
**Presentation:** 3
**Contribution:** 2
**Rating:** 5
**Confidence:** 3

**Summary:**

This pager introduces RLang-Dyna-Q, an extension of prior work, RLang, that transforms natural language into a formally specified language for solving tasks. Rather than focusing on policy-centric translations, as in much prior work, the authors observe that much of the advice or coaching offered by human experts will come in the form of statements that describe a transition function (e.g., "If you miss, you will fall down"), a reward function (e.g., "You get 10 points when you touch the key"), or a plan (e.g. "Go to X, then pick up Y, then return to Z"). RLang-Dyna-Q is a combination of Dyna-Q and RLang that uses the learned world model/transition functions of RLang to further refine the Q function learned in Dyna-Q.

The proposed RLang-Dyna-Q algorithm is compared to Dyna-Q and to a random policy in a handful of tabular RL domains, showing that it significantly outperforms Dyna-Q. The authors also perform an ablation study in which they test only subsets of RLang-Dyna-Q (only policy advice, only transition advice, or only plan advice).  Finally, the authors conduct a small user study in which 10 undergraduate students provide advice to warm start an RLang-Dyna-Q, with each student contributing one piece of advice, and 5/10 pieces of advice leading to policy improvements over the baseline, un-advised policy.


After reviewing the rebuttal, the authors have clarified the scope of the paper and their intended contributions and research area. Given the research direction and goals are more in line with language grounding and leveraging language for tasks, rather than improving task-learning performance or RL/IL efficiency, I will raise my contribution score to 2 and my overall score to 5.

**Strengths:**

* The paper is well written and provides a clear overview of the motivation, problem setting, and proposed solution.
* The paper proposes a blend of conventional planning literature and formal specification with the advancement of LLMs, leading to a significant improvement over conventional tabular RL solutions.
* The authors conduct a small scale user study, which solicits and leverages advice from untrained human coaches for a planning task.

**Weaknesses:**

* The method is not entirely clear, particularly given the heavy reliance on prior work (RLang) in this paper. It is not clear how the Q table relates to the RLang vocabulary files or RLang declarations, and this information must be obtained by referring to and reading prior work (meaning that RLang-Dyna-Q does not entirely stand on its own, but feels more like the "second half" of the RLang work).
* The results for RLang-Dyna-Q are not very convincing, and the comparison to a method that is nearly three decades old is also not very convincing. Comparisons to more modern RL baselines would improve the work. In particular, comparing to an LLM that generates Python/programming solutions seems like a very important baseline (even if there is no RL refinement, it would be useful to simply see to what extent an advanced LLM can solve these tabular domains out-of-the-box).
* The advice required to make RLang-Dyna-Q actually improve over baselines seems very particular. For example, looking at the advice in Figures 3-6, there is a combination of plans, general advice, and state transition advice. There is not a discussion or written analysis on what types of advice work best, or why. Similarly, the success of different types of advice seems extremely finicky. Comparing advice from participants 5 and 10 in the user study, the written advice is nearly identical. However, the performance deltas are quite significant (from a 33% increase to just a 2% increase).

**Questions:**

* Why not compare to conventional RL methods (e.g., PPO with a small neural network), to RLang itself, or to LLMs that generate code for plans?
* Why cut training off at 30-75 episodes, which is quite a small budget given that these are not expensive or safety-critical domains? It seems that one argument for RLang-Dyna-Q is that it could be is significantly more efficient than modern RL baselines by leveraging human advice, but if so then this should be shown by empirical comparisons (e.g., how many episodes does each method require to achieve maximum returns?).
* What differentiates good vs. bad advice for RLang-Dyna-Q? The user study provides great insight into the effects of different natural language prompts for the method. However, at times the prompts appear semantically identical, but they yield different results.

---

> ### Author Response · Authors · 2024-11-13
> **Part 1 of our Initial Response**
>
> We are grateful to the reviewer for their careful evaluation and helpful comments. Please find our responses to the critiques below.
>
> **Weaknesses**
>
> Regarding Weakness 1:
>
> Upon revisiting the method-related sections of the paper, we agree with your point that the paper heavily relies on the cited RLang paper, so we have added the following sentence at the end of section 3.1 to help explain to the reader how RLang-DynaQ works:
>
> > Similar to Dyna-Q, RLang-Dyna-Q leverages the Bellman update rule to update Q-values using rollouts collected both from environment interaction and from simulated interaction, which is generated from a partial model of the environment that is learned over time. However, RLang-Dyna-Q also leverages a partial model given by an RLang program to generate simulated rollouts before learning begins (see Algorithm \ref{alg:rlang-dynaq}, our modifications to Dyna-Q are in blue).
> >
>
> And also have added the following sentence at the end of section 3 before section 3.1 to explain what the original RLang work does:
>
> > These programs are compiled using RLang's compiler into Python functions corresponding to transition functions, reward functions, policies, and plans that can be leveraged by a learning agent.
> >
>
> We hope that these adjustments make this work feel more stand-alone.
>
> Regarding Weakness 2:
>
> We appreciate the reviewer’s comments on our usage of DynaQ and how we may compare our work to other baseline agents. However, we argue that DynaQ is reasonable tabular RL algorithm to base our RLang-enabled agent on and compare it to due to its simplicity and stature as an early discrete, model-based RL algorithm that was one of the first of its kind to learn from model-based simulated rollouts. Our goal in this work is to demonstrate how language can be used to inform a tabula rasa agent, and our choice of DynaQ was motivated by the simplicity of a discrete, tabular agent where various MDP components could more directly ground to in comparison to more modern deep learning methods in which integrating MDP components is less obvious. Integrating natural language advice into such deep RL algorithms is a pressing and interesting area that we leave open for future work.
>
> Regarding a comparison to an LLM that can generate a programming language-based solution, this is essentially how the RLang-DynaQ-Plan and RLang-DynaQ-Policy agents work. RLang's syntax for plans and policies is similar enough to Python for LLMs to understand it out of the box. Both of these agents defer to RLang plans and policies over learning itself---these agents will always choose to execute RLang plans and policies when they are applicable, performing essentially the same as an LLM+Python agent if the policies or plans are totally comprehensive. We again point out that our method is about integrating language advice into a reinforcement learning agent, *not* about maximizing agent performance. Under this framing, a comparison to LLM+Python is not very relevant.

---

> ### Author Response · Authors · 2024-11-13
> **Part 2 of our Initial Response**
>
> Regarding Weakness 3:
>
> Your point that the results of this method are somewhat finicky is well-taken. We have addressed some of your concerns by adding a paragraph at the end of section 4.2 to discuss how various kinds of advice impact the performance of agents:
>
> > The impact of each kind of advice (e.g. plans, policies, transitions, and rewards) varied across tasks in the VirtualHome and Minigrid experiments, with some environments benefiting primarily from plan-centric advice and others benefiting most from policy advice. In virtually all cases, model-centric advice---about transitions and rewards---was less valuable than other forms of advice. We suggest that this discrepancy is due to how useful model-based advice is in comparison to explicit policy and planning advice. While policy and planning advice describe which actions to take in a given context, model-based advice was often used to suggest which actions \textit{not} to take, relying on the underlying learning agent to find the best action. Furthermore, model-based advice was useful less of the time, i.e. in fewer states. This is best illustrated by comparing the relative performance of effect-enabled RLang-Dyna-Q agents with policy and plan-enabled agents in the MidMazeLava Experiment in Figure \ref{fig:midmazelava} and the FoodSafety Experiment in Figure \ref{fig:foodsafety}. The model-based advice in the first experiment is to avoid lava, which there are many opportunities to walk into, resulting in the performance of the effect-enabled agent closer to the plan and policy-enabled agents. By comparison, the model-based advice in the second experiment is more niche, accounting only for a handful of transitions, and the effect-enabled agent correspondingly performs closer to baseline Dyna-Q than to the plan and policy-enabled agents.
> >
>
> We note that this has bumped the user study table to the appendix.
>
> Regarding your specific comparison of the 5th and 10th piece of advice in the user study, we note that the advices are semantically different in a crucial sense: that the 10th piece of advice makes no mention of the grey door, which must be opened before going to the red door, while the 5th piece of advice explicitly addresses opening the grey door. We agree that this seems like a small difference, but when grounding the advice to an executable plan it makes a meaningful difference. We have included the following sentence at the end of section 4.4 to address your concern about the final piece of user advice:
>
> > Failures also occurred when users specified plans whose pre-conditions were not met at the start state of the environment and failed to execute (e.g. the last piece of advice suggests to go to the room with the red key, but the agent cannot visit the room without first opening the grey door).
> >
>
> **Questions**
>
> Regarding Question 1:
>
> We don’t compare to conventional RL methods, RLang on its own, or LLMs because the goal of this work is to propose a method for leveraging human language advice that can be used to improve the performance of RL agents. Specifically, we demonstrate how advice about various components of MDPs---including reward functions, transition functions, policies, and plans---can be integrated comprehensively into a single model-based RL agent. We compare our agent (RLang-DynaQ) to a structurally-identical agent (DynaQ) that does not use language advice. We don’t claim to perform competitively against modern deep RL methods, as the center of our work is on language grounding, not maximizing agent performance. We believe this work would be valuable to the language-informed RL community.
>
> Regarding Question 2:
>
> We cut training off after a small number of episodes because they are sufficient to demonstrating that language-informed agents can learn faster than uninformed agents. Language advice can be extremely potent, and its effects on performance can be seen immediately in most cases. We note that the reward charts used in this paper do not plot average episodic reward on the Y-axis, they plot **cumulative reward**, i.e., when the reward curves achieve a stable slope (not a slope of 0), it means the agents have converged, yielding the same amount of reward on each time step. In nearly all experiments we run all agents until they plateau, i.e. when cumulative reward has reached a stable slope.
>
> Regarding Question 3:
>
> We address this question in our response to Weakness 3.
>
> We again thank the reviewer for their comments and questions, and welcome any additional feedback.

---

> > ### Author Response · Authors · 2024-11-21
> >
> > We appreciate the time and effort you have taken to provide feedback on our submission. We have worked diligently to address your comments and concerns and believe the revisions we've made as a result significantly improve the paper. If our responses and updates resolve your questions, we kindly ask you to consider revisiting your score. Please do not hesitate to let us know if you have additional feedback or concerns.

---

> > > ### Author Response · Authors · 2024-11-29
> > >
> > > I wanted to kindly follow up on my earlier response to your review. We’ve worked hard to address your comments and would greatly appreciate any further feedback you may have.

---

### Official Review · Reviewer_q6RU · 2024-11-10

**Soundness:** 3
**Presentation:** 3
**Contribution:** 2
**Rating:** 5
**Confidence:** 4

**Summary:**

The paper proposes a framework to leverage natural language-based advice to accelerate RL learning process. The RLang-Dyna-Q algorithm extends the original RLang framework and combines it with the traditional Dyna-Q. Empirical results over two sets of experiments help verify the effectiveness of propose algorithms.

**Strengths:**

1. The writing is overall good and easy to follow.
2. The idea of translating natural language advice to RLang and using RLang to generate synthetic transitions makes sense.
3. The writing flow in the experiment is great – sec 4.1 and 4.2 present two effective cases with assumptions on semantically-meaningful labels, while sec 4.3 also presents efforts to try to address this assumption. Also, user study has been completed in Table 2.

**Weaknesses:**

1. Only Q-learning-based RL is tested in the experiment. More advanced and modern RL algorithms are needed to show the generality, e.g. PPO.
2. More LLM + RL baselines are needed. There are a few simple alternatives to directly leverage LLM to process natural language advice to help RL training. For example, what if we don’t use any RLang-based program, and only treat LLM’s as the generator for actions and transitions?
3. Another important assumption (and limitation) in the paper is that each environment will be provided with human-annotated natural language advice. This is a strong prior compared with all RL baselines. The author needs to discuss more about this assumption and whether we can use any other ways to bypass the need for human labels. For example, could LLMs directly generate advice for any given environment?
4. More qualitative results are needed for section 4.3 (a demo is not enough)

**Questions:**

1. Any idea why the DynaQ baseline doesn’t work in Figure 6’s experiment?
2. Typo in line 164.
3. If we are going to extend the algorithm to high-dimensional continuous RL problem, what could be the biggest challenges?

---

> ### Author Response · Authors · 2024-11-13
>
> We appreciate the reviewer’s insightful feedback and constructive suggestions. We have addressed their concerns below.
>
> **Weaknesses**
>
> Regarding Weakness 1:
>
> The reviewer correctly points out that we only use Q-Learning-based RL methods in this work. Our goal in this work, however, is to demonstrate how language can be used to inform a tabula rasa agent, and our choice of DynaQ was motivated by the fact that by choosing a discrete tabular agent, we were better able to isolate the impacts of our language grounding approach on learning in comparison to more modern deep learning methods. Integrating natural language advice into such deep RL algorithms is a pressing and interesting area that we leave open for future work.
>
> Regarding Weakness 2:
>
> We appreciate the reviewer’s suggestions about including LLM + RL baselines, but we point out that the suggested baseline the reviewer discusses is essentially how the RLang-DynaQ-Plan agent works—in this case, RLang acts as an intermediary to convert action plans into a policy that can be run by the RL agent. Regarding comparisons to other agents, the goal of this work is to demonstrate that grounding language to every component of an MDP can improve performance, and our experiments demonstrate this. We don’t claim that this is the most efficient way to ground language, and we welcome follow-up works that can ground language without using RLang.
>
> Regarding Weakness 3:
>
> The topics the reviewer mentions have been the subject of existing works, but the focus of our work is precisely on how to ground human-given language in RL. The assumption of language for an environment is part of the problem statement we aim to solve, and not a limitation.
>
> Regarding Weakness 4:
>
> We believe that the symbol-grounding via VLM demonstration in section 4.3 shows that VLMs can be used to obviate the need for some human annotations. The symbol-grounding problem is outside of the scope of this work, but we invite the reviewer to elaborate on what kind of qualitative results would aid a reader in finding this method convincing.
>
> **Questions**
>
> Regarding Question 1:
>
> This is a good question, similar to one that was raised by Reviewer 3iuj. We believe DynaQ performs poorly in this environment due to non-determinism in the VirtualHome simulator. This is not an intended feature of the environment. We note this in the description for Figure 6.
>
> Regarding Question 2:
>
> Typo fixed, thanks!
>
> Regarding Question 3:
>
> This is an important question and we thank the reviewer for raising it. Our approach relies on RLang, a formal symbolic language, for capturing language advice. While natural language itself is symbolic and discrete, MDPs may not be, and this raises an important question on how to bridge the gap between a symbolic syntax and a continuous semantics, or meaning. One possible solution could be to represent relevant symbolic action and perception abstractions with continuous functions. For example, representing a “Kick” skill with a Dynamic Movement Primitive or a “is_open()” predicate with a Convolutional Neural Network. This question suggests many open avenues for future work.
>
> We again thank the reviewer for their questions and comments, and welcome any additional feedback.

---

> > ### Author Response · Authors · 2024-11-21
> >
> > We appreciate the time and effort you have taken to provide feedback on our submission. We have worked diligently to address your comments and concerns and believe the revisions we've made as a result significantly improve the paper. If our responses and updates resolve your questions, we kindly ask you to consider revisiting your score. Please do not hesitate to let us know if you have additional feedback or concerns.

---

> > > ### Comment · Reviewer_q6RU · 2024-11-28
> > >
> > > Thank you for the detailed rebuttal and clarifications. I appreciate the authors addressing my concerns thoughtfully.
> > >
> > > I still believe the experimental scope is not comprehensive enough to fully validate the framework's effectiveness across diverse RL paradigms and environments. While I understand the authors’ intent to focus on grounding language in a discrete tabular setting, demonstrating the generality of the proposed approach would require more diverse experimental setups.
> > >
> > > Moreover, the qualitative results in section 4.3 indeed can show an example of it, but it doesn't present metrics like its grounding accuracy, how integrating the system into RL can influence the final performance, etc. That's why the quantitative results are necessary.
> > >
> > > For these reasons, I will maintain my original score,

---

> > > > ### Author Response · Authors · 2024-11-30
> > > >
> > > > Thank you for the feedback. Can you elaborate more on the experimental setups that you believe readers would want to see to be convinced, given that we are grounding language to a discrete tabular setting? We have evaluated our method on over half a dozen environments. Furthermore we are able to ground information to every element of the MDP---to our knowledge no other work has done this---and introduced a new tabular RL agent for language grounding based on DynaQ. The symbol-grounding demonstration in sec 4.3 shows that human annotated grounding files (which must only be written once for an entire class of environments) can be automated in part by VLMs. VLMs do not need to be directly integrated into the pipeline, and we don't a priori expect VLMs to do all the symbol-grounding work for us. Quantitative results for such a demonstration are somewhat tangential to the thrust of our work, as it would essentially be a separate evaluation of the capabilities of VLMs. We do report quantitative results for the LLM groundings with a user study, however, which has been moved to the appendix.
> > > >
> > > > Unfortunately, we are unable to make edits to the paper this late in the review process. We believe there is a language+RL community at ICLR that would be interested in this approach to grounding language, and who would find our methodology and demonstration using numerous experiments convincing. We humbly ask you to consider this when making your final decision.

---

### Author Response · Authors · 2024-12-03
**Summary of Review and Rebuttal Process for ACs and SACs**

Some in the community have suggested that a concise summary of reviews and rebuttals would assist reviewers, ACs, and SACs during the next part of the review process. We have tried our best to summarize the main strengths and weaknesses mentioned in our reviews, as well as our responses to them.

In terms of readability, all of the reviewers remarked that the paper was well-written, easy to follow, well-contextualized, and well-motivated, with one praising the writing flow in the experiments section in particular (reviewer q6RU, score: 5) and another stating, “This is an important research topic and the contribution is contextualized nicely” (reviewer 3iuj, score: 6). Some of the reviewers were impressed with the quantity, variety, and quality of experiments performed (reviewer 3iuj, score: 6), with one reviewer highlighting our “extensive experiments” (reviewer 8geU, score: 6), which included experiments done across two sets of environments (6 total environments with our agent, multiple ablations, and a baseline), a user study, a two-tiered symbol-grounding demo using a VLM, and an additional experiment comparing directly to an additional baseline (SayCan). Others were unimpressed by the experiments, with one citing DynaQ as a poor point of comparison that is “nearly three decades old” and suggesting a comparison to a deep-learning based agent and an LLM without any RL (reviewer EByn, score: 3). Our work, however, is precisely on how to integrate language into RL, and our language-informed RL agent called RLang-DynaQ is based on DynaQ, which, due to its simplicity, is a prime candidate for a clear and minimal example showing how language can impact performance in classical model-based RL. Another stated that our VLM demonstration—an addendum demo designed to show that a VLM can replace the need for human-generated labels by using a VLM to label objects directly—needed more quantitative results, “a demo is not enough” (reviewer q6RU, score: 5). We believe a demo is sufficient to show this relatively straightforward image-labeling capability.

Overall we found the reviews to be very constructive, and made a substantial number of updates to the paper based on helpful feedback. This included an additional full paragraph elucidating the methodology (suggested by reviewer EByn), another full paragraph dissecting the impact of various kinds of advice on agent performance (suggested by reviewers EByn and 3iuj), a sentence elaborating on how advice was collected in our user study (suggested by reviewer 3iuj), and we increased the font size on all reward plots (suggested by reviewer 3iuj). All reviewers responded positively to our updates, with reviewer 3iuj increasing their contribution from fair (2) to good (3). While most reviewers responded to our rebuttals, the author of our paper’s most critical review (EByn, score: 3), whose review we responded to in detail on November 12th (the day that reviews were released), has not yet responded to our rebuttal at the time of this comment. We note that the average score reported for all reviewers who engaged with us in the rebuttal process is (6+6+5)/3 = 5.666.

---

### Meta-Review · Area_Chair_c7Qb · 2024-12-21

**Metareview:**

The paper introduces RLang-Dyna-Q, an extension of the RLang framework, which grounds natural language advice in a formal language (RLang) and integrates it into the Dyna-Q algorithm to improve reinforcement learning (RL). The method leverages human-provided language to enhance the learning of RL agents, particularly in transition and reward functions. Empirical results show improvements in RL tasks using natural language advice from human experts.

Reasons to accept
- The idea of grounding natural language advice to improve RL performance is novel and well-motivated.
- The paper presents multiple experiments across domains like Minigrid/BabyAI and VirtualHome, demonstrating the benefits of the proposed approach.
- The inclusion of a user study, where human participants provide advice, offers valuable qualitative insights into how different types of language inputs influence performance.
- The method works towards reducing the reliance on dense human annotations, which shows potential for real-world application.

Reasons to reject
- The experiments primarily focus on Q-learning, which is considered somewhat outdated. Comparisons to more advanced and widely used algorithms, such as SAC and PPO, are missing.
- Performance improvements are tied to specific types of advice, and it is not clear which types of advice are most effective or how to generalize them across tasks.
- A major limitation is the assumption that high-quality advice must be provided by human experts.
- The reliance on in-context learning for language grounding may restrict the model's ability to adapt and refine its understanding. Making the language model trainable alongside the RL agent could yield better results.
- The method's connection to prior work (RLang) is not fully clear, and the paper requires readers to reference previous work for a complete understanding. The algorithm is presented as an extension rather than a standalone solution.

While this paper studies a promising research direction and presents an interesting approach, I believe its weaknesses outweigh its strengths. Consequently, I recommend rejecting the paper.

**Additional Comments On Reviewer Discussion:**

During the rebuttal period, three reviewers acknowledged the author's rebuttal.

---

### Decision · Program_Chairs · 2025-01-22

Reject